# Perinatal mortality in German dairy cattle: Unveiling the importance of cow-level risk factors and their interactions using a multifaceted modelling approach

Yury Zablotski[1]*, Katja Voigt[1], Martina Hoedemaker[2], Kerstin E. Müller[3], Laura Kellermann[1], Heidi Arndt[2], Maria Volkmann[4], Linda Dachrodt[2], Annegret Stock[3]

1 Faculty of Veterinary Medicine, Clinic for Ruminants with Ambulatory and Herd Health Services, Ludwig-Maximilians-Universität München, München, Germany, 2 Clinic for Cattle, University of Veterinary Medicine Hannover, Hannover, Germany, 3 Faculty of Veterinary Medicine, Clinic for Ruminants, Freie Universität Berlin, Berlin, Germany, 4 Faculty of Veterinary Medicine, Institute for Veterinary Epidemiology and Biostatistics, Freie Universität Berlin, Berlin, Germany

* Y.Zablotski@med.vetmed.uni-muenchen.de

**Data Availability Statement:** All data files will be available from the Mendeley Data: https://data.mendeley.com/datasets/kcwh6p384f/1.

## Abstract

Perinatal mortality (PM) is a common issue on dairy farms, leading to calf losses and increased farming costs. The current knowledge about PM in dairy cattle is, however, limited and previous studies lack comparability. The topic has also primarily been studied in Holstein-Friesian cows and closely related breeds, while other dairy breeds have been largely ignored. Different data collection techniques, definitions of PM, studied variables and statistical approaches further limit the comparability and interpretation of previous studies. This article aims to investigate the factors contributing to PM in two underexplored breeds, Simmental (SIM) and Brown Swiss (BS), while comparing them to German Holstein on German farms, and to employ various modelling techniques to enhance comparability to other studies, and to determine if different statistical methods yield consistent results. A total of 133,942 calving records from 131,657 cows on 721 German farms were analyzed. Amongst these, the proportion of PM (defined as stillbirth or death up to 48 hours of age) was 6.1%. Univariable and multivariable mixed-effects logistic regressions, random forest and multimodel inference via brute-force model selection approaches were used to evaluate risk factors on the individual animal level. Although the balanced random forest did not incorporate the random effect, it yielded results similar to those of the mixed-effect model. The brute-force approach surpassed the widely adopted backwards variable selection method and represented a combination of strengths: it accounted for the random effect similar to mixed-effects regression and generated a variable importance plot similar to random forest. The difficulty of calving, breed and parity of the cow were found to be the most important factors, followed by farm size and season. Additionally, four significant interactions amongst predictors were identified: breed—calving ease, breed—season, parity—season and calving ease—farm size. The combination of factors, such as secondiparous SIM breed on small farms and experiencing easy calving in summer, showed the lowest probability of PM. Conversely, primiparous GH cows on large farms with difficult calving in winter exhibited the highest

**Funding:** This study was funded by the German Federal Ministry of Food and Agriculture, via the Federal Office for Agriculture and Food (https://www.ble.de/EN/Home/home_node.html), grant numbers 2814HS006, 2814HS007 and 2814HS008. Martina Hoedemaker received those fundings. The funders had no role in study design, data collection and analysis, decision to publish, or preparation of the manuscript.

**Competing interests:** The authors have declared that no competing interests exist.

probability of PM. In order to reduce PM, appropriate management of dystocia, optimal heifer management and a wider use of SIM in dairy production are possible ways forward. It is also important that future studies are conducted to identify farm-specific contributors to higher PM on large farms.

# 1 Introduction

Despite significant research interest, perinatal calf mortality (PM) remains an important issue in the dairy farming industry as shown by a systematic review and meta-analysis of the subject, which revealed an increased incidence risk of PM over time [1]. High perinatal mortality results in the loss of animals, financial loss [1], and raises concerns about animal welfare [1], which is unacceptable from both ethical and economic perspectives. The implications of various health conditions leading to peri- and neonatal mortality on the welfare of the individual calf have been reviewed by Mellor and Stafford [2]. While these authors rank stillbirth as less severe than death caused by postnatal diseases from an individual welfare perspective [2], pain associated with a difficult calving experienced by the dam [3] should also always be taken into account when discussing the welfare implications of stillbirth rates. Despite the importance of PM in these aspects and despite a significant number of publications on the subject, the identification of universally applicable predisposing factors remains difficult because of a high variation in case definitions or farm characteristics. This lack of universal and comparable data is also most likely a hindrance to the development of improvement schemes, and a lack of programmes addressing calf PM has been highlighted by Compton and co-workers [1]. A major difficulty for the comparability of studies is caused by an inconsistent definition of PM [1,4,5]. Specifically, authors often include the death of a full-term calf either during (i.e. stillbirth) or shortly after parturition, but this post-natal period varies between one [6] and 48 [7] hours. There is also variation as to whether abortions are included in this definition [5]. Santman-Berends and co-workers [4], for instance, proposed one of the broadest definitions of PM by including aborted calves (>100 days in gestation), stillbirths and any postnatal mortality before the time of ear-tagging (usually at the age of three days, but this varies from farm to farm). Such different inclusion criteria greatly hinder the comparability of results, since even slight differences in definitions of PM can produce very different results [4]. A standardized definition, proposed by Wong and co-workers [8], includes all calves that are stillborn or die within 48 hours of birth over the total number of still- and live-born calves, and the authors suggest this definition should be applied in any future studies on the topic. According to a literature review, the estimates of average PM in dairy calves (based on varying definitions) ranged from 2.4 to 9.7% across the 26 reviewed studies, the majority of which were conducted on the animal level [5]. However, there is a lack of agreement amongst studies regarding the risk factors associated with PM. Therefore, reducing PM without a clear understanding of these risk factors is impossible. Previous attempts to aggregate the current knowledge on PM in dairy cattle have been made, but further studies using large data sets and a consistent definition of PM are necessary to fully understand this issue. In addition to the high variability in case definitions, two comprehensive reviews of the international literature [5,9] also highlighted further challenges for comparison and interpretation of the results of various studies due to the high variability in methods of data collection, farm characteristics and locations, inconsistent inclusion of PM predictors, and the use of highly variable or simple statistical methods for analyzing multivariable data in an attempt to describe complex relationships. Additionally, the

description of statistical methods in some of the studies was found to be incomprehensible [10,11].

For example, using logistic regression without inclusion of random effects to study PM as a binomial response variable is appropriate for data from a single farm [7,12], but less suitable for multi-farm data sets. This is because farms often differ in characteristics such as herd size or climate, and farm-specific variance cannot be assessed or avoided without inclusion of random effects, leading to results that may not accurately reflect reality. Using mixed-effects logistic regression for multi-farm (or multi-herd) data seems to be a more reasonable approach, but only few previous studies have used this method [13–15]. Additionally, the complexity of statistical tests varied greatly amongst the studies [5]. Only univariable tests were used by Bleul [16], while other authors applied single (bivariate) interactions between predictors [17] or multiple regressions without interactions [6,18], while multiple regressions with interactions were only rarely applied [12–15].

Another limitation of the published literature on PM in dairy cattle is that the vast majority of studies frequently focus on the Holstein-Friesian breed, thus limiting their applicability to other breeds. Only some previous studies, such as Bleul [16], have included a variety of dairy and beef breeds in their examination, but they have not investigated potential differences between individual dairy breeds. This lack of examination thus masks potential differences in PM between Holstein-Friesian cattle and other common dairy or dual-purpose breeds used in the dairy industry, such as Brown Swiss or German Simmental [19]. Particularly the German Simmental breed is widely used in dairy production in Southern Germany, for instance accounting for 77.4% of dairy cattle in the large federal state of Bavaria, with Brown Swiss making up another 11.2% of the dairy animals in this state (n = 908,265 dairy cows contributing to the Bavarian milk recording database [20]). In contrast, German Holstein and related breeds are predominantly used in Northern and Eastern Germany [21]. According to the breeding associations, over 1.1 million Simmental cows are used in dairy production in Germany [19]. However, previous studies examining PM in German dairy cattle have only focused on medium to large-scale dairy units in the federal states of Saxony and Thuringia, operating with Holstein-Friesian and closely related breeds (red Holstein, blackpied dairy cattle, and their crosses). These studies are further limited by only including an individual farm [18] or a relatively small number of dairy units [22]. Under these circumstances, a PM risk of 9.7% and 9.3%, respectively, has been reported on the examined German dairy farms [18,22].

The current study was therefore undertaken to (1) further evaluate PM in dairy cattle in Germany on the cow-level while exploring potential breed differences between German Holstein, Simmental and Brown Swiss cattle, (2) to contribute to the clarification and resolution of contradictions in previous studies by exploring risk factors and interactions that potentially influence PM and (3) to increase comparability amongst studies for PM by providing results of several statistical techniques. We hypothesize that breed differences in PM risk exist and that the association between PM and potential risk factors are also breed dependent. Additionally, we propose that an ensemble of different statistical techniques would not yield identical outcomes but rather unveil the most pivotal risk elements for PM.

## 2 Material and methods

### 2.1 Data collection and editing

Data were collected between 2015 and 2019 as part of an extensive cross-sectional study across Germany, which had been initiated and funded by the German Federal Ministry of Food and Agriculture, via the Federal Office for Agriculture and Food, grant numbers 2814HS006, 2814HS007 and 2814HS008. This initial project aimed to achieve a comprehensive and

representative description of animal health on German dairy farms, and to develop options for potential stakeholder action in relevant animal health-related areas [21]. Within Germany, three major geographical regions were assessed: North, to include the federal states of Lower Saxony and Schleswig-Holstein; East, including Mecklenburg-Western Pomerania, Thuringia, Saxony-Anhalt and Brandenburg; and South, represented by the federal state of Bavaria.

Considering that the target variables for health damage are exclusively quantitative in nature (e.g., percentage of animals in a herd with clinical mastitis) and that the risk factors are both quantitative and qualitative in nature, the sample size should be calculated to be appropriate for different distribution scenarios. To calculate an optimal and affordable sample size, different possible distribution scenarios were calculated with a power of 80% and a significance level of 5%. For example, a standard deviation of 7 and an accuracy of ±1, ±2, ±3, and ±4 were assumed for estimating an expected value [23].

Sample size calculation was based on the formula suggested for prevalence studies:

$$n = \frac{Z^2 P (1 - P)}{d^2}$$

where $n$ is the sample size to be calculated, $Z$ the level of confidence, $P$ the assumed prevalence, and $d$ the precision (calculations according to Glaser and Kreienbrock [23].

The described calculations suggested an optimal and achievable sample size of 250 farms per region [21,23,24]. Farm selection was stratified by administrative district and herd size, i.e. the number of lactating and dry cows, within the federal states and study regions.

The national animal information database (HIT, regions North and East) as well as the milk recording database (Milchprüfring Bayern e.V., region South) provided the basis for farm selection. The regional differences in the databases used were caused by differences in co-operation by the respective regional veterinary authorities, which acted as a point of contact for farms drawn from the HIT database [21]. Farms were randomly drawn from these databases using an automated approach. Within each study region, a total of 1,250 farms, i.e. 5 times the number required for the study, were initially drawn from the underlying population to cover a response rate of at least 20%. Region specific herd size cut-off values were determined to obtain a realistic distribution of herd sizes within the study population, and to account for regional structural differences in dairy farming in Germany. The farms were approached in writing by the organizations with access rights to the farmers' personal details as previously described [21]. Participation in the study was voluntary, and prospective participants contacted the regional study coordinators to express their interest following their invitation. Since the initial response rate fell below expectations, further farms had to be drawn from the databases to reach the target sample size. Of all invited farms, participation of 5.9% (260 / 4,418; region South), 9.1% (253 / 2,787; region North) and 14.5% (252 / 1,739; region East) could be achieved [21]. The University Animal Welfare Representative (University of Veterinary Medicine, Hanover, Germany) was contacted, and verbal consent was obtained that ethical review and approval was not required, because at the time this work was planned and conducted, prospective approval of this research by an animal or human research ethics committee was not required in Germany [25]. The reason was that this study did not contain any animal experiments which required any approval by the animal health and welfare authorities, and that informed consent was obtained in writing from interested farm managers for participation, data inspection and publication. All farm-specific information was handled confidentially according to the principles of German and European data protection legislation.

A total of 30 researchers (region North: 10; region East: 13, region South: 7) were involved in the various aspects of the underlying comprehensive study [21] and visited a total number of 765 farms on a single occasion between December 2015 and August 2019. All researchers

were qualified veterinary surgeons and had been previously trained by the study co-ordinators to ensure a standardized approach to data collection across all regions. Different farms were thus visited in different years and different seasons. The primary focus of these visits was the assessment of health parameters for adult dairy cattle and various young stock age groups, as well as husbandry, nutrition and milk yield. Retrospective calving records were made available by the participants to cover the 12-month period prior to the farm visit. All calving records therefore constituted farmer-recorded data. The calving records were made available by the farmers to the primary research group [21]. The current study was later undertaken to make use of this data set. Certain limitations to data collection (such as, for instance, a lack of data on corresponding insemination dates) therefore had to be accepted, since data acquisition could not prospectively be planned to serve the specific purposes of this study, and only covered a 12-month period for each farm. Production level (e.g. milk yield parameters) and cow-related information (e.g. calving date, calving ease, calf mortality parameters) was retrieved from the national animal recording (HIT) and from the respective regional milk recording systems (Dairy Herd Improvement, DHI). The available data included: date of farm visit, farm size, breed, ear tag number, parity, calving date, calving ease, stillbirth or death within the first 48 hours of life, calf sex and number of calves per birth (singletons or multiples). These data were imported from the national cattle registration and milk recording databases for the initial study. The information from both sources was subsequently merged and underwent extensive plausibility checks by the initial researchers as previously described [21]. In brief, the Institute for Biometry, Epidemiology and Information Processing, University of Veterinary Medicine, Hanover, Germany, conducted general plausibility checks on approximately 230 quantitative variables from the interview questionnaires and approximately 50 variables from the data collection sheets. These checks included automated consistency checks, analysis for logical relationships, and manual review of discrepancies between data sources (HIT and DHI). By doing this, the team achieved a very low number of missing values and high data quality through collaboration between scientists from different regions and extensive data review [21].

Forty-four of the 765 participating farms did not have any available data from the milk recording database (DHI) and were thus excluded from the current analysis. Our initial data set thus contained information from 721 farms. Perinatal mortality was defined as all calves that were stillborn at term or died within 48 hours (2 days) of birth over of the total number of still- and live-born calves, following the suggestion by Wong and co-workers [8]. Data were recorded by the farmers as part of their routine herd records in the mentioned databases, which, amongst other information, require entry of the birth of full term calves or calf mortality, but not abortions. The classification as stillbirth at term or postnatal death within the first two days of life is thus based on the farmers' assessment. Exact data on gestational length, crown-rump length or the exact hour of postnatal death was not available to verify these records.

The data set for individual animals was checked for completeness, and a total of 13,345 missing values for calving ease out of the 133,942 parturition records (9.96%) were imputed via a non-parametric multivariable imputation approach by chained random forest algorithm with 1000 trees [26]. This method combines random forest imputation [27,28] with predictive mean matching [29] and thus iterates multiple times until the average out-of-bag (OOB) prediction error of predictive models stops to improve. The OOB prediction error for calving ease stopped improving near 0, indicating very accurate predictions for the 9.96% of missing records regarding the course of parturition. Thus, imputations for the predictor "calving ease" were accepted to be included in the analysis. Similar imputation of the variable calf sex, which had 8,745 (6.53%) missing values, was rejected due to a high (47%) OOB prediction error. Moreover, the majority (84.5%) of missing values for calf sex fell upon calves recorded as

stillborn. This parameter could therefore not be used as a valid predictor for PM. Our data showed that 3.1% (n = 4,100) of all parturitions were identified as multiples (twins or triplets). However, PM was recorded towards only 2.1% of these multiple births, which seems unrealistically low. Besides, the available dataset did not provide any information on how many of the calves within one multiple litter experienced PM or survived, or on the sex of those stillborn or surviving calves from multiple litters. Thus, both variables, calf sex and number of calves per birth, could not be used as potential predictors for PM in our study. Only single births were therefore included in the analysis. Similarly, corresponding artificial insemination dates were not available for the vast majority of parturitions. Calculation of gestation length was therefore difficult, and became too uncertain as a potential predictor to be included into the analysis.

Following exclusion of twin or triplet births for the above reasons, the final dataset was based on 133,942 calving records of 131,657 cows from 721 farms from the three regions, and the following predictors were available for statistical analysis of PM: farm size, breed, parity, calving ease and season. There are no universally accepted categories for farm size, and what constitutes a small, medium sized or large farm very much depends on typical farm sizes in a studied country or region. To avoid the application of arbitrary categories, the numerical variable farm size was therefore binned (categorized) into three categories (small, medium, large) by the "quantile" method. Binning was meant to reduce any bias by splitting the farm size variable (minimum = 1 animal, maximum = 2,821, mean = 162, Standard Deviation (SD) = 243.6, median = 79, Interquartile range (IQR) = 135) into approximately equal groups and resulted in the following categories: 257 small farms (1–52 animals), 253 medium sized (53–129 animals) and 255 large farms (130–2,821 animals). Three main breeds were identified: German Holstein (GH, including red Holstein), Brown Swiss (BS) and Simmental (SIM). Seventeen other regional or rarer breeds and crossbred animals were categorized as "other".

Parity was categorized as primiparous ($1^{st}$ lactation), secondiparous ($2^{nd}$ lactation) and multiparous ($\geq 3^{rd}$ lactation). The difficulty of calving (calving ease) was recorded by the farmers and assigned to three categories: easy, medium (one helper and light use of mechanical tools) or difficult (several helpers, mechanical pulling tools, or surgery, e.g. caesarean section or fetotomy). Season was extracted from the calving date and was assigned to the categories winter (December 15th to March 14th), spring (March 15th to June 14th), summer (June 15th to September 14th), and autumn (September 15th to December 14th).

## 2.2 Statistical analyses

All analyses were conducted using R Statistical software (R version 4.0.3, 2020; RStudio desktop version 1.4.1103, 2021). All packages used in the current study are listed in the supplementary material (Table 6.1 in S1 File).

Perinatal mortality was studied as the response variable. Initially, potential predictors such as region, season, parity, calving ease, breed, and farm size were considered. However, in regions North and East, 94% and 93% of the cows were GH, and 5% and 7% were "others", respectively, while there were only very few SIM or BS (both rounded to 0%). In contrast, 80% of the dairy cattle in region South were SIM and 9% BS, with only 8% GH and 3% other breeds. The variables "breed" and "region" were thus found to be highly multicollinear (i.e. to contain very similar information as measured by Variance Inflation Factor (VIF)). Such multicollinearity resulted in the exclusion of the variable "region" as a potential predictor for analysis, although differences in PM between regions North and East (each with $\geq$93% GH) may potentially exist. VIFs in multivariable models without interactions were checked, and all examined predictors showed VIF < 1.5, which indicated no multicollinearity issues in our final model (i.e. no remaining redundant predictors).

Logistic mixed effects models were used to predict perinatal mortality with season, parity, calving ease, breed and farm size. All five predictors were analysed on the animal level (1) using five univariable models, (2) the multivariable model without interactions, (3) ten separate bivariable pairwise interactions models and (3) the final model with multiple, only relevant interactions. All models included individual farm as random effect fitted on the intercept. Since the vast majority (98.4%) of cows only contributed a single parturition to the dataset, inclusion of "individual cow" as a random effect did not improve the quality of the model and was therefore rejected. Models with random slopes did fail to converge or, if converged, often reduced the quality of the model. Thus, random intercept-only mixed effects regressions (using BOBYQA optimizer) were modeled. Backwards stepwise elimination with an inclusion criterion of a *P value* < 0.05 was applied to variables as well as interactions in order to (1) control for all possible risk factors and to reduce the number of variables (or interactions) to only potentially influential ones, while (2) maximizing model quality at the same time. Equations and further details of the modelling approach are shown in supplemental materials.

Models were compared amongst each other using five main performance quality indicators: Akaike's Information Criterion (AIC), Bayesian Information Criterion (BIC), conditional coefficient of determination $R^2$, marginal coefficient of determination $R^2$ and the intraclass-correlation coefficient (ICC). The model showing the best combination of predictive (AIC and BIC) and fitting power (conditional $R^2$, marginal $R^2$, ICC) was preferred.

Contrasts between particular categories of variables were assessed after model-fitting with the Benjamini & Hochberg *P value* correction for multiple comparisons [30]. In all our models, we applied proportional weights to the different categories based on their frequencies in the original data. This approach helps to address the issue of unbalanced data, where certain groups may have more observations than others. Results with a *P value* < 0.05 were considered statistically significant, while results with a *P value* < 0.1 were considered suggestive. The 95% confidence intervals were chosen as a measure of uncertainty for all metrics.

The importance of variables and their pairwise interactions in the multivariable model was determined via random forest algorithm, which allowed for the comparison of variables using the mean decrease accuracy (MDA). The variables with the highest MDA give the best prediction and thus contribute the most to the model [31]. The validation of the final model was conducted via splitting data into a training (80%) and testing dataset (20%), training the model, predicting the PM of the testing dataset and comparing the predicted with known mortality data. The binary classification of PM for the two strata, "surviving past 48 hours of life, no PM" and "stillbirth or death within 48 hours *post natum*, PM", was carried out by sampling 1000 from each stratum at every tree. Such stratification is necessary because if sampling is not controlled for strata, the hypothetical 95% of calves surviving past 48 hours could affect the model to predict mostly "no PM" and thus achieve high accuracy, but skew the variable importance.

A novel *"brute force"—automated model selection and multimodel inference* approach was applied to validate the choice of mixed-effect and random forest models. This approach, implemented in the "glmulti" statistical package [32], finds the best set of models amongst all possible models. Specifically, we evaluated 32 models resulting from various combinations of five predictors and an additional 1450 models generated from all possible pairwise interactions between these predictors. The "brute-force" approach ranked these models based on their Akaike Information Criterion (AIC) values, identifying the optimal model or a set of qualitatively similar models within a 2-unit range of AIC. Notably, this approach allowed us to identify the best-performing model, characterized by four interactions (as depicted in Fig 3). The corresponding model equation is provided in the supplementary materials. Subsequently, we compared this model's AIC with that of the "final" model obtained through a backwards-

selection approach, aiming to contrast the two variable selection methodologies. Similarly to the random-forest model, the brute-force approach also allows to measure the importance of predictors. The importance value for a particular predictor or interaction is equal to the sum of the weights for the models in which the variable appears [32]. Thus, a variable that appears in many models with large weights will receive a high importance value.

The importance of cow-level risk factors associated with PM and interactions amongst those risk factors were purposefully studied with four different methods in this study in order to validate those methods against each other, uncover potential variations in results depending on the method used, and in order to increase the comparability of the current study to other publications. The latter is necessary because the sole use of univariable models is common but rarely justified in previous studies, as they ignore the influence of potentially confounding factors. Complex methods, such as multivariable models with multiple interactions, are often avoided due to small data sets or due to difficulties in interpretation. Additionally, this study compares all results (acquired with mixed-effects logistic regressions) to the logistic regression (estimated in parallel) to investigate whether logistic regression produces different estimates, and mis-represents reality in the multi-farm environment. The study also employs variable importance techniques, such as random forest and multimodel inference via brute-force model selection approach, to rank the predictive power of predictors and their pairwise interactions for PM.

## 3 Results

A total of 133,942 calving records for single calves from 131,657 cows were analyzed. The overall prevalence of PM was found to be 6.1% (n = 8,211). When averaged over the 721 separate farms, the mean prevalence (values in brackets: standard deviation, SD) in our study was 5.2% (3.8) while the median prevalence (values in brackets: IQR = Q25%—Q75%) was 5.3% (5.2 = 2.3–7.5). Regarding breed, the majority (85.5%) of the studied population were GH (n = 114,466), followed by SIM accounting for 7.3% (n = 9,836) and the BS breed (0.9%; n = 1,254). "Other" breeds accounted for 6.3% of the cases (n = 8,386). In terms of parity, 33.5% of the cows were primiparous (n = 44,868), while 24.4% (n = 32,640) were secondiparous, and the remaining 42.1% (n = 56,434) were multiparous. Out of a total of 133,942 cases, 83.1% (n = 111,364) experienced easy calving, 13.6% (n = 18,207) encountered medium difficulty, and 3.3% (n = 4,371) faced a difficult parturition. The farm size distribution showed that 5.7% of the cases (n = 7,632) were from small farms, 16.0% (n = 21,398) from medium-sized, and the majority, comprising 78.3% (n = 104,912), from large farms. Finally, the distribution of seasons indicated that 26.0% (n = 34,751) of the calves were born in summer, followed by autumn and winter with 25.9% (n = 34,708) and 25.2% (n = 33,799) respectively, while spring accounted for 22.9% (n = 30,684) of the parturitions. Table 1 summarizes the number and percentage of perinatal deaths associated with the studied parameters.

### 3.1 Results of univariable and multivariable analyses without interactions

The univariable analyses showed calving ease, parity, breed and farm size to be significantly (all *P values* < 0.001) associated with PM, while season was not significant (Fig 1A, *P value* = 0.28). In the consecutive multivariable model without interactions, again, calving ease, parity, breed (all *P values* < 0.001) and farm size (*P value* = 0.04) remained significant, while only season did not show any significant relationship with PM (*P value* = 0.4, Fig 1B). The Chi-Square statistics, used as the effect size for mixed-effect logistic regressions (Fig 1A and 1B), and mean decrease accuracy, used as the effect size for random forest (Fig 1C), both ranked the importance of the five predictors identically, showing calving ease and parity to be the most influential predictors for PM, with calving ease showing a much higher importance for PM compared

**Table 1. Description of variables included in the statistical analyses as potential risk factors for perinatal mortality of 133,942 single calves born between December 2015 and August 2019 on 721 German dairy farms.**

| Variable | Live calves, n = 125,731[1] | Perinatal death, n = 8,211[1] |
|---|---|---|
| **Breed** | | |
| German Holstein | 107,156 (93.6%) | 7,310 (6.4%) |
| Simmental | 9,453 (96.1%) | 383 (3.9%) |
| Brown Swiss | 1,184 (94.4%) | 70 (5.6%) |
| others | 7,938 (94.7%) | 448 (5.3%) |
| **Parity** | | |
| 1 | 40,779 (90.9%) | 4,089 (9.1%) |
| 2 | 31,311 (95.9%) | 1,329 (4.1%) |
| 3+ | 53,641 (95.1%) | 2,793 (4.9%) |
| **Calving ease** | | |
| easy | 106,685 (95.8%) | 4,679 (4.2%) |
| medium | 16,346 (89.8%) | 1,861 (10.2%) |
| difficult | 2,700 (61.8%) | 1,671 (38.2%) |
| **Binned farm size [N°]** | | |
| small [1 – 52] | 7,303 (95.7%) | 329 (4.3%) |
| medium [53–129] | 20,285 (94.8%) | 1,113 (5.2%) |
| large [130–2821] | 98,143 (93.5%) | 6,769 (6.5%) |
| **Season** | | |
| summer | 32,687 (94.1%) | 2,064 (5.9%) |
| autumn | 32,588 (93.9%) | 2,120 (6.1%) |
| winter | 31,671 (93.7%) | 2,128 (6.3%) |
| spring | 28,785 (93.8%) | 1,899 (6.2%) |

[1]n (%).

to parity. The brute-force model selection approach ranked parity as the most influential predictor, closely followed by calving ease and breed (Fig 1D). All four approaches found season to have the least importance for PM incidence, if no interactions were included.

The results of univariable models are included in Table 2 in order to facilitate the comparability with other studies. However, since the influence of a particular predictor needs to be adjusted for the influence of the other predictors, only results of the multivariable models were interpreted. Such adjustment for every particular predictor was made by averaging the results for this particular predictor over the levels of the remaining ones. For instance, PM incidence varied across breeds. The SIM breed showed the lowest (3.0%), the BS breed the highest (4.9%) predicted incidence of PM. The odds of PM of SIM calves were lower than GH, BS and others (detailed results and *P* values are shown in Table 2). The PM incidence also varied across calving difficulty levels. The lowest (3.2%) predicted incidence was found when calving is easy, as opposed to predicted 10.0% for medium difficulty and 37.0% for difficult parturitions. All contrasts in odds of PM amongst pairwise calving ease categories were significant. The predicted PM incidence was lowest for small farms (3.2%) as compared to large farms (4.2%), while medium size farms (with a predicted incidence of 3.9%) did not differ from small and large farms (Table 2). Primiparous cows showed the highest predicted PM incidence (5.9%), which differed substantially from secondiparous (3%) and multiparous (3.6%) animals. Interestingly, the predicted incidence of PM was lower in secondiparous cows than in multiparous cattle. When averaged over the levels of remaining predictors, similar predicted incidences of PM of around 4% were found across all seasons.

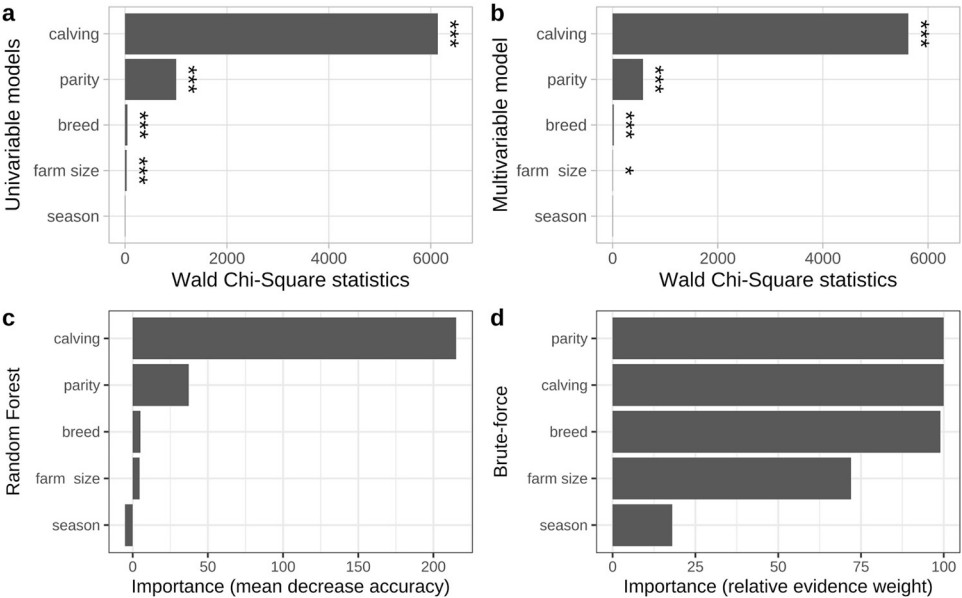

**Fig 1.** The importance of variables for the prediction of perinatal mortality measured by: the univariable (a) and multivariable (b) Analysis of Deviance Table (Type III Wald Chi-Square tests) with global P values, by the random forest algorithm via mean decrease accuracy (c) and by the brute-force model selection tool (d) which measures the importance value for a particular predictor by the sum of the weights for the models in which the predictor appears during the variable selection procedure. Significance codes: '***' < 0.001, '**' < 0.01, '*' < 0.05, '.' < 0.1'. 'Calving ease' is abbreviated as 'calving' to conserve plot space. All analyses were based on 133,942 parturitions from 721 farms.

If asked for predictions of particular combinations of levels of variables (e.g. breed has levels "GH", "SIM", "BS" and "other"), our multivariable model revealed the highest probabilities of PM, ranging from 39.1% for SIM to 52.3% for BS, for **primiparous cows on large farms, experiencing a difficult calving in winter**. Similarly, the most favorable conditions for a birth, and thus the lowest probability of PM, were found to be in **secondiparous cows on small farms, having an easy calving in summer**, with SIM again showing the lowest predicted incidence of PM amongst breeds of 1.3% under these conditions, and BS again showing the highest PM risk (2.2%), while GH (1.9%) and others (1.8%) showed an intermediate risk. (R code for predicted incidences of all scenarios is shown in supplementary materials).

## 3.2 Results of analyses with single and multiple interactions

While seven of the separately modeled pairwise interactions were found to be significantly associated with PM and one showed a tendency (breed—calving ease *P value* < 0.001, calving ease—farm size *P value* < 0.001, parity—season *P value* < 0.001, breed—season *P value* = 0.041, breed—parity *P value* = 0.027, parity—farm size *P value* = 0.013, breed—farm size *P value* = 0.079; Fig 2A), only four were found to be important by the multivariable model with interactions (Fig 2B). Particularly, three of the most influential interactions showed a very strong relationship with PM, namely breed—calving ease (*P value* < 0.001), parity—season (*P value* < 0.001) and calving ease—farm size (*P value* < 0.002), while the fourth, breed—season (*P value* < 0.036, Fig 2B) showed a weaker, but still present association. The brute-force method agreed with the multivariable model (Fig 2D). The random forest method, however, agreed with the first three interactions (parity—season, breed—calving ease and calving ease—farm size), but ranked the breed—parity interaction as the fourth most important interaction, closely followed by breed—season.

**Table 2. Results of univariable and multivariable mixed-effects logistic regressions on the cow-level and pairwise comparison of potential risk factors for perinatal mortality of 133,942 single calves born between December 2015 and August 2019 on 721 German dairy farms.** Odds ratios (ORs) are represented by the ratio of odds of one category in the numerator (e.g., SIM) to the odds of another category in the denominator (e.g., GH). The specific category in the numerator serves as the reference category for this particular OR.

| Predictor (pairwise comparison) | Univariable Models | | | Multivariable Model | | |
|---|---|---|---|---|---|---|
| | OR[1][2] | 95% CI[2] | *P value* | OR[2][3] | 95% CI[2] | *P value* |
| **Breed** | | | | | | |
| SIM / GH | 0.63*** | 0.52, 0.77 | <**0.001** | 0.69*** | 0.54, 0.89 | <**0.001** |
| BS / GH | 0.95 | 0.64, 1.41 | 0.986 | 1.17 | 0.75, 1.84 | 0.794 |
| BS / SIM | 1.50 | 0.98, 2.29 | **0.072** | 1.70* | 1.07, 2.71 | **0.017** |
| others / GH | 0.85* | 0.73, 0.98 | **0.015** | 0.95 | 0.82, 1.11 | 0.842 |
| others / SIM | 1.33** | 1.06, 1.68 | **0.007** | 1.38* | 1.04, 1.83 | **0.016** |
| others / BS | 0.89 | 0.59, 1.36 | 0.896 | 0.81 | 0.51, 1.29 | 0.655 |
| **Calving ease** | | | | | | |
| medium / easy | 3.66*** | 3.39, 3.96 | <**0.001** | 3.37*** | 3.12, 3.64 | <**0.001** |
| difficult / easy | 19.1*** | 17.4, 20.9 | <**0.001** | 17.7*** | 16.2, 19.4 | <**0.001** |
| difficult / medium | 5.21*** | 4.71, 5.77 | <**0.001** | 5.26*** | 4.75, 5.83 | <**0.001** |
| **Farm size** | | | | | | |
| medium / small | 1.20 | 0.98, 1.47 | **0.082** | 1.23 | 0.96, 1.56 | 0.119 |
| large / small | 1.53*** | 1.27, 1.85 | <**0.001** | 1.31* | 1.02, 1.69 | **0.033** |
| large / medium | 1.27*** | 1.10, 1.48 | <**0.001** | 1.07 | 0.89, 1.29 | 0.650 |
| **Parity** | | | | | | |
| 2 / 1 | 0.42*** | 0.39, 0.46 | <**0.001** | 0.51*** | 0.47, 0.55 | <**0.001** |
| (3+) / 1 | 0.53*** | 0.49, 0.56 | <**0.001** | 0.60*** | 0.56, 0.64 | <**0.001** |
| (3+) / 2 | 1.24*** | 1.15, 1.35 | <**0.001** | 1.18*** | 1.09, 1.28 | <**0.001** |
| **Season** | | | | | | |
| autumn / summer | 1.03 | 0.95, 1.12 | 0.781 | 1.04 | 0.95, 1.13 | 0.720 |
| winter / summer | 1.06 | 0.98, 1.16 | 0.211 | 1.06 | 0.97, 1.15 | 0.378 |
| winter / autumn | 1.03 | 0.95, 1.12 | 0.747 | 1.02 | 0.94, 1.11 | 0.945 |
| spring / summer | 1.04 | 0.96, 1.13 | 0.621 | 1.01 | 0.93, 1.11 | 0.978 |
| spring / autumn | 1.01 | 0.93, 1.10 | 0.991 | 0.98 | 0.90, 1.07 | 0.927 |
| spring / winter | 0.98 | 0.90, 1.06 | 0.904 | 0.96 | 0.88, 1.05 | 0.652 |

[1]*p<0.05

**p<0.01

***p<0.001.

[2]OR = Odds Ratio, CI = Confidence Interval.

[3]SIM = Simmental, GH = German Holstein, BS = Brown Swiss.

## 3.3 Interaction between breed and calving ease

The interaction between breed and calving ease became the most revealing and showed the widest range of predicted probabilities of PM, with the lowest probability predicted for easy calving ranging from 2.8% in the breed category "others" to 4.4% in BS, and the highest predicted incidence for difficult calving ranging from 15.1% in SIM to 41.7% in "others". Calves from difficult parturitions had a greater risk of PM than calves from medium or easy calvings in all studied breeds (Fig 3A and Table 3). While GH and "other" breeds had a lower risk of PM at easy calving as compared to medium calving difficulty, the risk of PM for SIM and BS in easy and medium difficulty calvings did not differ. The risk of PM for calves from SIM cows with difficult and medium difficulty calving was lower than for GH and others (Fig 3A and

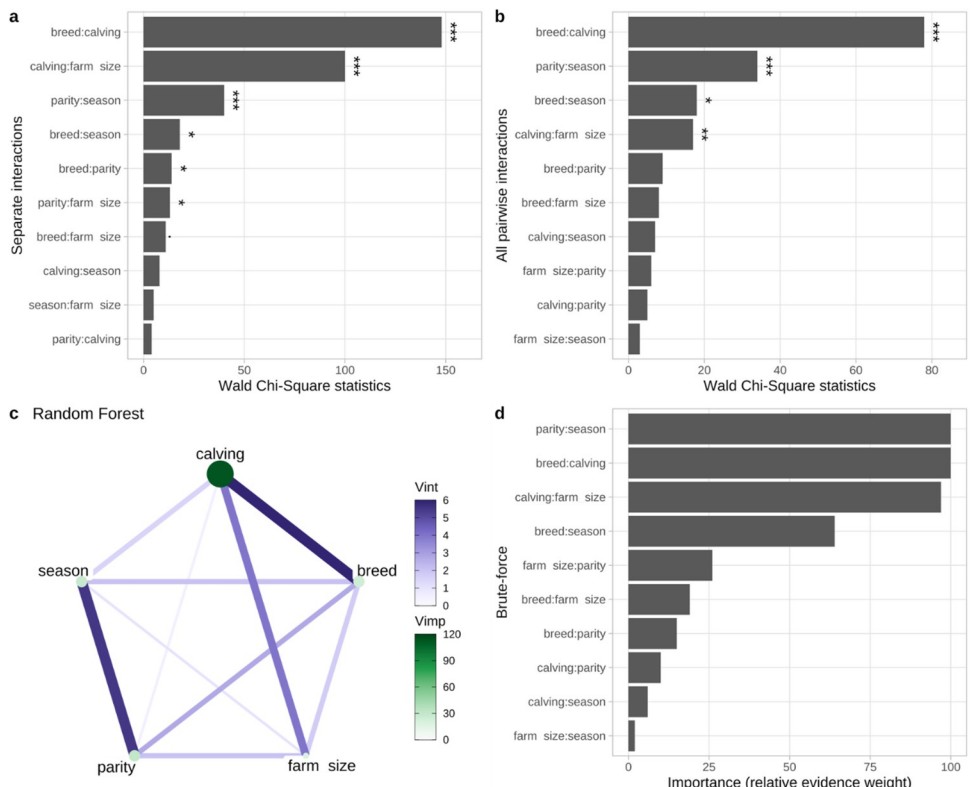

**Fig 2.** The importance of all pairwise interactions for the prediction of perinatal mortality measured by: the separate interactions (a) and all interactions in the same model (b) via Analysis of Deviance Table (Type III Wald Chi-Square tests) with global P values, by the random forest algorithm via mean decrease accuracy (c) and by the brute-force variable selection tool (d), which measures the importance value for a particular predictor by the sum of the weights for the models in which the predictor appears during the variable selection procedure. "Vint" stands for the importance of interactions and "Vimp" stands for the importance of variables (predictors). Significance codes: '***' < 0.001, '**' < 0.01, '*' < 0.05, '.' < 0.1'. 'Calving ease' is abbreviated as 'calving' to conserve plot space.

Table 4). The SIM breed also showed the smallest range in PM risk across calving categories, ranging from 3.4% for easy to 15.1% for difficult calving. In contrast, "other" breeds showed the widest range in PM risk across calving categories, ranging from 2.8% for easy to 41.7% for difficult parturitions.

### 3.4 Interaction between parity and season

Primiparous cows showed the highest (5.4% - 6.9%), while secondiparous showed the lowest (2.9% - 3.5%) risk of calf PM across seasons (Fig 3B). Nearly all parities within any season differed significantly from each other. Details and *P* values are given in Table 4. The only exceptions were observed for autumn and spring, where the PM risk for calves from secondiparous cows (3.4%) did not differ from the multiparous category (parity = "3+"). Primiparous cows also showed the widest variance across seasons. Namely, winter (6.9%) had a higher predicted incidence of PM compared to all other seasons, while summer (5.4%) had the lowest risk compared to all other seasons. The risk of PM was similar in primiparous cows in autumn and spring. We did not find any differences between seasons in parities greater than 1. Further details and all *P* values are shown in Table 3.

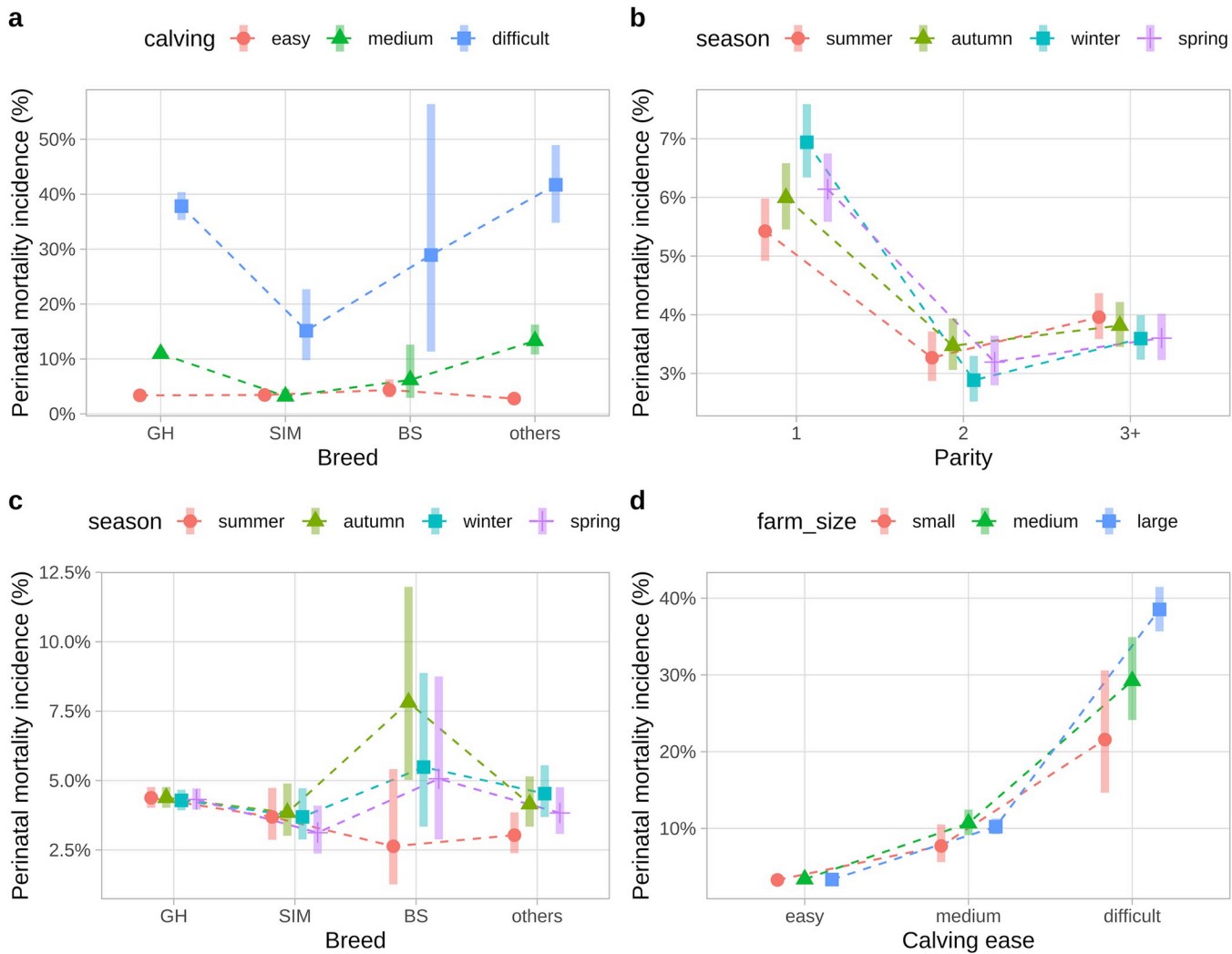

**Fig 3.** Probabilities of perinatal mortality predicted by the four most important interactions which remained after several interaction selection procedures in the final mixed-effects logistic regression: breed—calving ease (a), parity—season (b), breed—season (c) and calving ease—farm size (d). Breed categories: GH–German-Holstein, SIM–Simmental, BS–Brown-Swiss, others–multiple breeds with low counts. Parity: 1 –primiparous, 2 –secondiparous, 3+—multiparous. Calving ease: easy—no assistance, medium—one helper and light use of mechanical tools, or difficult—several helpers, mechanical pulling tools, or surgery. 'Calving ease' is abbreviated as 'calving' to conserve plot space.

## 3.5 Interaction between breed and season

The interaction between breed and season became significant (*P value* = 0.04) due to the BS breed, which showed the widest range of probabilities of PM incidence, ranging from the smallest predicted incidence of 2.6% in summer to the highest incidence of 7.8% in autumn (Fig 3C). The BS breed, however, had the smallest number of observations (average of 314 calvings per season) and therefore produced the widest confidence intervals, indicating the highest uncertainty. In contrast, the GH breed, accounting for the majority of the data, showed the narrowest confidence intervals. GH cows calving in summer had higher odds of PM than "others" in summer (*P value* = 0.013, Table 4). The PM risk of the BS breed in autumn was higher compared to all other breeds (GH, SIM and others, all *P values* = 0.021) in autumn (Table 4 and Fig 3C). The predicted incidence of PM of calves born to BS cows in summer was lower

**Table 3. Only significant contrasts extracted from the four most influential interactions by mixed-effects logistic regression for potential risk factors for perinatal mortality of 133,942 single calves born between December 2015 and August 2019 on 721 German dairy farms.** Odds ratios (ORs) are represented by the ratio of odds of one category in the numerator (e.g., easy) to the odds of another category in the denominator (e.g., medium). The specific category in the numerator serves as the reference category for that particular OR.

| predictor | contrast | OR [95% CI][b] | P value |
|---|---|---|---|
| breed—GH[a] | calving ease—easy / medium | 0.28 [0.26–0.31] | <0.001 |
| breed—GH | calving ease—easy / difficult | 0.06 [0.05–0.06] | <0.001 |
| breed—GH | calving ease—medium / difficult | 0.2 [0.18–0.23] | <0.001 |
| breed—SIM[a] | calving ease—easy / difficult | 0.2 [0.11–0.37] | <0.001 |
| breed—SIM | calving ease—medium / difficult | 0.18 [0.09–0.38] | <0.001 |
| breed—BS[a] | calving ease—easy / difficult | 0.11 [0.03–0.47] | 0.001 |
| breed—BS | calving ease—medium / difficult | 0.16 [0.03–0.84] | 0.012 |
| breed—others | calving ease—easy / medium | 0.19 [0.14–0.26] | <0.001 |
| breed—others | calving ease—easy / difficult | 0.04 [0.03–0.06] | <0.001 |
| breed—others | calving ease—medium / difficult | 0.21 [0.14–0.33] | <0.001 |
| parity—1 | season—summer / autumn | 0.9 [0.79–1.03] | 0.05 |
| parity—1 | season—summer / winter | 0.77 [0.67–0.88] | <0.001 |
| parity—1 | season—summer / spring | 0.88 [0.76–1.01] | 0.017 |
| parity—1 | season—autumn / winter | 0.86 [0.75–0.97] | 0.004 |
| parity—1 | season—winter / spring | 1.14 [1–1.3] | 0.014 |
| breed—BS | season—summer / autumn | 0.32 [0.11–0.96] | 0.036 |
| breed—others | season—summer / winter | 0.66 [0.44–0.98] | 0.036 |
| calving—difficult | farm size—small / large | 0.44 [0.24–0.81] | 0.004 |
| calving—difficult | farm size—medium / large | 0.66 [0.46–0.95] | 0.009 |

[a]SIM = Simmental, GH = German Holstein, BS = Brown Swiss.

[b]OR = odds ratio, CI = confidence intervals.

than in autumn (*P value* = 0.036, Table 3). The odds of PM of "other" breeds in summer was lower than in winter (*P value* = 0.036, Table 3).

## 3.6 Interaction between farm size and calving ease

The interactions between farm size and calving ease revealed that difficulty of calving was much more important than farm size. Particularly, the predicted probabilities of PM ranged from 21.6% to 38.5% for difficult calvings on various farm sizes, while they were very similar around 3.3% at any farm size when calving was easy. When calving was of medium difficulty, they ranged from 7.7% to 10.7% across farm sizes (Fig 3D). Difficult parturitions had significantly higher odds of PM as compared to medium and easy calving on all farm sizes. Detailed results and *P* values are given in Table 4. Similarly, medium calving difficulty showed a higher risk of PM compared to easy calving across all farm sizes. Thus, the interaction between farm size and calving ease only showed a significant association with PM due to the difficult calvings, for which large farms had higher odds of PM compared to small and medium sized farms (Table 3), while farm size did not significantly matter when calving was easy or of medium difficulty.

## 4 Discussion

### 4.1 Key findings of modelling framework

Our final mixed-effect model with interactions revealed a wide range of PM risk, highly dependent on specific combinations of predictor levels. The model with interactions provided

**Table 4. Only significant contrasts extracted from the four most influential interactions by mixed-effects logistic regression for potential risk factors for perinatal mortality of 133,942 single calves born between December 2015 and August 2019 on 721 German dairy farms.** Odds ratios (ORs) are represented by the ratio of odds of one category in the numerator (e.g., GH) to the odds of another category in the denominator (e.g., others). The specific category in the numerator serves as the reference category for that particular OR.

| predictor | contrast | OR [95% CI][b] | P value |
|---|---|---|---|
| calving—easy | breed—GH / others[a] | 1.22 [1–1.48] | 0.053 |
| calving—easy | breed—BS / others[a] | 1.59 [0.93–2.74] | 0.071 |
| calving—medium | breed—GH / SIM | 3.73 [2.27–6.14] | <0.001 |
| calving—medium | breed—GH / others | 0.8 [0.59–1.1] | 0.095 |
| calving—medium | breed—SIM / others[a] | 0.22 [0.12–0.38] | <0.001 |
| calving—medium | breed—BS / others | 0.43 [0.14–1.28] | 0.082 |
| calving—difficult | breed—GH / SIM | 3.41 [1.7–6.84] | <0.001 |
| calving—difficult | breed—SIM / others | 0.25 [0.11–0.54] | <0.001 |
| season—summer | parity—1 / 2 | 1.7 [1.44–2] | <0.001 |
| season—summer | parity—1 / (3+) | 1.39 [1.23–1.58] | <0.001 |
| season—summer | parity—2 / (3+) | 0.82 [0.7–0.96] | 0.004 |
| season—autumn | parity—1 / 2 | 1.77 [1.51–2.08] | <0.001 |
| season—autumn | parity—1 / (3+) | 1.61 [1.42–1.82] | <0.001 |
| season—winter | parity—1 / 2 | 2.51 [2.13–2.96] | <0.001 |
| season—winter | parity—1 / (3+) | 2 [1.76–2.27] | <0.001 |
| season—winter | parity—2 / (3+) | 0.8 [0.67–0.95] | 0.002 |
| season—spring | parity—1 / 2 | 1.98 [1.68–2.34] | <0.001 |
| season—spring | parity—1 / (3+) | 1.75 [1.53–2.01] | <0.001 |
| season—spring | parity—2 / (3+) | 0.88 [0.74–1.05] | 0.092 |
| season—summer | breed—GH / others | 1.46 [1.05–2.03] | 0.013 |
| season—autumn | breed—GH / BS | 0.54 [0.29–1.02] | 0.021 |
| season—autumn | breed—SIM / BS | 0.47 [0.24–0.93] | 0.021 |
| season—autumn | breed—BS / others | 1.96 [0.98–3.92] | 0.021 |
| farm size—small | calving ease—easy / medium | 0.41 [0.26–0.63] | <0.001 |
| farm size—small | calving ease—easy / difficult | 0.12 [0.07–0.22] | <0.001 |
| farm size—small | calving ease—medium / difficult | 0.3 [0.15–0.6] | <0.001 |
| farm size—medium | calving ease—easy / medium | 0.29 [0.24–0.36] | <0.001 |
| farm size—medium | calving ease—easy / difficult | 0.08 [0.06–0.12] | <0.001 |
| farm size—medium | calving ease—medium / difficult | 0.29 [0.2–0.41] | <0.001 |
| farm size—large | calving ease—easy / medium | 0.3 [0.28–0.33] | <0.001 |
| farm size—large | calving ease—easy / difficult | 0.06 [0.05–0.06] | <0.001 |
| farm size—large | calving ease—medium / difficult | 0.18 [0.16–0.21] | <0.001 |

[a]SIM = Simmental, GH = German Holstein, BS = Brown Swiss.
[b]OR = odds ratio, CI = confidence intervals.

deeper insights into the combinations of risk factors compared to the model without interactions, as it allowed modelling any level of one predictor within any level of any other predictor. The results of modern machine learning approaches, such as random forest and brute-force variable selection, were congruent with those obtained from the application of a traditional mixed-effect model. This convergence not only supports the validity of the classical modelling approach, but also highlights the promising potential of machine learning methods in advancing the fields of agriculture and veterinary sciences.

The overall PM rate in our dataset (6.1%) was consistent with the mean perinatal mortality of 6.2% reported in a comprehensive literature review by Cuttance and Laven [5], but lower

than the results of previous studies from Germany, which reported a PM rate on the individual animal level of 9.5% (1,417 / 14,920) and 9.7% (45 / 463), respectively [18,22]. High PM rates have also been reported from Canada (9.6% (9,048 / 94,250), [33]) and Denmark (9% (160,352 / 1,781,694), [34]), while other studies reported lower rates, particularly from Norway (2% (10,569 / 528,475), [35]) and Switzerland (2.4% (50,932 / 2,122,184), [16]). Mee and co-workers reported a PM risk for the Holstein-Friesian breed ranging from 3.5 to 12.1% depending on the country of origin [13].

The observed discrepancy in predicted PM risk between the various studies may in part be attributed to the choice of statistical methods. Specifically, the disparity between univariable (e.g. [16]) and multivariable analyses highlights the importance of conducting the latter. While reporting results from univariable models enhances comparability with other studies, it is essential to recognize that they may overlook crucial information regarding potential confounding effects. Whenever feasible, the application of multivariable models is therefore advisable.

The second potential source of variability in predicted PM risk estimates between studies may partially originate from the utilization of less sophisticated models when dealing with intricate data characterized by a hierarchical structure (grouped, clustered, or dependent). The use of logistic regression has been widely adopted in studies investigating risk factors for PM on the cow level [5]. However, while this method may be adequate for short-term data collection or single farm studies (e.g. [12]), it may be insufficient for data collected over longer time periods or multi-farm studies, as these provide additional variation in data from the repeated measurements of both the individual animal, if applicable, and the farm. In such cases, the use of mixed-effects logistic regression, incorporating random effects for either or both animal and farm, is a more appropriate statistical approach. Our findings demonstrate that the quality of logistic regression models applied to the same data is consistently inferior to mixed-effects models, since logistic regression models tend to overestimate the risk of PM. The fact that most previous studies utilized logistic regression suggests that the current understanding of PM may be inflated or overstated when applied to large datasets from multiple farm environments. For instance, the goodness of fit of the multivariable logistic model without interactions, as measured by the Tjur $R^2$ (0.077), was found to be three times inferior compared to the goodness of fit of the mixed model, as measured by the conditional $R^2$ (0.223) and marginal $R^2$ (0.126) in our study. These results highlight the need for a more nuanced approach to understanding PM risk in larger and more complex data environments.

Variability in PM estimates may also arise from the complexity of the models applied. Specifically, while a multivariable model without interactions delivers reliable results, models with interactions may reveal a much wider range of PM estimates. These interaction models thus provide deeper insights into the combinations of risk factors.

The utilization of modern machine learning methods, such as random forest classifier in a multi-farm environment requires caution as its implementation can prove hazardous, for example if samples are not balanced for an unequal distribution of data. For instance, the majority of the calves in our dataset survived past 48 hours (93.9%), while the minority (6.1%) experienced PM. A balanced random forest is achieved when each tree within the ensemble contains an equal number of surviving and dead calves. Specifically, this balance is attained by down-sampling the majority class (survived) and over-sampling the minority class (dead) during the bootstrapping iterations [36]. Both a balanced random forest algorithm and the multivariable model without interactions determined that calving ease and parity were the two most influential predictors, while the predictor breed was placed third and showed substantially less importance as compared to calving and parity. However, when samples were not balanced, a random forest ranked the importance of breed second with slightly higher importance as

compared to parity. The unbalanced random forest can thus mislead researchers by yielding a seemingly high predictive accuracy. However, such false accuracy is often driven by an over-representation of the majority class (in this case, surviving calves). For example, if PM is 6%, and the random forest algorithm predicts 100% survival, the reported accuracy would be 94%. Nevertheless, this result is fundamentally flawed due to the class imbalance. The balanced random forest model without random effect produced identical results to mixed-effects and brute-force methods without interactions and very similar results to models with interactions and is thus deemed a valuable tool in determining variable and interaction importance.

The brute-force approach represents an advancement over the commonly used backwards or forwards variable selection methods, as it does not entail a sequential reduction or addition of variables, but rather compares all models with all possible combinations of predictors (or interactions) to determine the optimal model that provides the most accurate description of the data. Given the frequent disparity between "optimal" final models produced by forwards and backwards variable selections, it is highly recommended to employ the brute-force approach whenever feasible. Furthermore, the brute-force approach generates variable (and interaction) importance plots, akin to those produced by random forest, but based on differing decision criteria. This not only validates the random forest approach, but also provides an alternative perspective, which can only serve to enhance the inference produced from data.

## 4.2 Breed differences

In contrast to the well-studied Holstein-Friesian breed, dual-purpose breeds such as Simmental and traditional Brown Swiss cattle have received limited attention when studying dairy calf mortality parameters. While some breed comparisons have been made in the past, such as by Voljč et al. [37], who reported a lower perinatal and neonatal mortality risk in calves born to Simmental, Brown Swiss and other breeds as compared to Holstein-Friesian dams, our results partly corroborate these findings: an important aspect of our study is the observation of a lower PM risk in Simmental cows in comparison to the German Holstein breed, whereas the results for Brown Swiss cattle did not differ significantly from the PM risk for any of the other three breed categories. The low sample size of BS cattle in our study led to high uncertainty and widely overlapping confidence bands.

Our results, in conjunction with the study by Voljč et al. [37], therefore suggest a potential for utilizing Simmental cattle bred for dual-purpose use such as German Simmental more widely in dairy production in order to improve welfare by reducing calf losses. Lower mortality is desirable from both ethical and economic perspectives, and dual-purpose breeds like Simmental present a positive contrast to the high mortality rates observed in German Holstein cattle. The Simmental breed shows good potential for dairy production, with an average 305-day milk yield of 8080 kg and good average milk solids (fat: 4.23%; protein: 3.54%; n = 701,775) as shown by Bavarian milk recording data [20]. While this is lower than the average 305-day milk yield of German Holstein cattle kept in the same federal state (n = 80,682; milk yield: 9,445 kg; fat: 4.14%; protein: 3.44%) [20], the simultaneous high potential for meat production lends a high slaughter value to both cull cows and youngstock [19], making German Simmental cattle an economically viable breed in dairy farming. The underlying causes for the observed breed difference in PM risk, however, require further investigation. It is very likely that the Simmental breed is truly more robust, and that this robustness leads to increased calf survival. However, it is also plausible that SIM owners might invest considerably higher efforts into calving management and postnatal care compared to GH owners due to the higher value associated with SIM calves, given their dual-purpose nature and high slaughter value. Although it is strongly discouraged to engage in such practices, in reality, male GH calves are sometimes

neglected and left to perish rather than investing in their treatment [38]. In this context, it would have been highly desirable to assess calf sex as a potential predictor for PM, as male calves of purely dairy breeds carry a particularly low economic value and are often regarded as a "by-product" of the dairy industry [39]. The high percentage of missing values in the available dataset regarding the sex of calves experiencing perinatal mortality, however, precluded this analysis in the current study. Further observational studies are therefore desirable to assess this important topic.

### 4.3 Season

The challenge of comparing the impact of season on PM across studies is rooted in several causes. Firstly, only a limited number of studies have evaluated the potential effect of season on PM. Additionally, the reported impact of season on PM is highly inconsistent across previous publications, and logistic regression analysis was applied in the vast majority of these [5]. This reference to methods is again noteworthy, as these findings suggest that the significant results observed in previous studies may have been exaggerated by the use of logistic regression and large sample sizes (such as 182,026 calvings [13]), both of which tend to increase the prominence of any observed effects.

The inconsistent results of previous studies, and the only significant effect of season when interacting with two other variables in our own study suggest that season may not be an important predictor of PM in dairy cattle. This finding is supported by the results of two variable importance analysis methods used in our study, which ranked "season" as the least important predictor of PM. It is therefore unlikely that moving to seasonal calving patterns would have any impact on reducing perinatal mortality on dairy farms.

### 4.4 Calving ease

Cuttance and Laven [5] reported that only eleven of 26 reviewed studies specifically examined the relationship between calving ease and PM. Despite variations in the definition and categorization of "assistance at calving" amongst these studies, all reported a significant association between calving ease and PM. In our study, calving ease emerged as the most significant predictor of perinatal mortality (PM) when analyzed using mixed-effects models, random forest analysis and the brute-force variable selection approach. This finding is in agreement with previous research by Chassagne et al. [40], who also identified calving ease as the primary cause of PM.

Two studies identified a significant interaction between calving ease and parity, with higher odds of perinatal mortality for calves born to young cows as compared to older cattle [13,41]. Our own study also observed a similar pattern, although the interaction was not statistically significant, as the pattern was observed across all categories of calving difficulty. However, our model revealed two further statistically relevant interactions. Namely, the effect of calving ease on PM is affected by breed and farm size. The highest probability of PM for difficult calvings was observed on large farms, potentially due to limited availability of personnel for assistance at calving, or a possibly lower personal interest of staff in calving outcomes in contrast to smaller, family-run farms. The Simmental breed was found to have the lowest PM risk amongst the various breeds, even in instances of high calving difficulty. It is worth noting that Simmental cattle were predominantly kept on smaller farms, where there may be fewer calving events and thus more involvement from the farmer, potentially resulting in higher calf survival rates. A farm categorized as "large" and keeping Simmental cattle is not necessarily of the same size as a large farm keeping German Holstein cattle, and is more likely to be placed at the lower end within the "large" farm size category.

The high importance of difficult calvings and, thus, dystocia for perinatal mortality risk provides several opportunities for reduction of these losses. These include further focus on calving ease as a criterion for breeding selection, the appropriate choice of sires, and optimal management and observation of late pregnant cows, leading up to parturition management that focuses on calf survival. It is likely that staff training and offering rewards for live-born calves might help reduce PM particularly on larger farms. Surgical intervention had only been recorded towards 0.12% of all studied parturitions. Further encouraging the use of caesarean section in the interest of calf survival by giving surgical delivery priority over forced vaginal delivery attempts and a sometimes possibly excessive use of mechanical pulling devices is also likely to reduce perinatal mortality rates.

## 4.5 Parity

Many previous studies revealed that calves born to primiparous cows had higher odds of perinatal mortality than calves born to multiparous animals [7,12–14,22,35,41–43]. However, Mee et al. [13] and Meyer and co-workers [41] did not find parity to be a standalone significant predictor of PM, but rather identified two significant interactions: parity and the degree of calving assistance and parity and the sex of the calf. Our model showed similar results with regard to higher odds of PM in primiparous cattle, but we did not observe a significant interaction between parity and calving ease, and were unable to explore any potential interactions between parity and calf sex due to missing data.

Interestingly, our model predicted a pattern similar to that found by Bicalho et al. [44], where the probability of PM declined in secondiparous cows as compared to primiparous dams, but then increased again in multiparous animals. This may be explained by the importance of dystocia / calving ease on PM. The incidence of dystocia is generally higher in heifers than in cows [45]. The risk of dystocia is, however, also influenced by a great variety of other factors, such as, for instance, cow body condition, calf birth weight, and any underlying disorders in the dam such as, for instance, metabolic diseases or vaginal prolapse [45]. It is highly likely that the combined risk for dystocia considering all potential influential factors is lowest and thus most favourable for calf survival in secondiparous cattle. For instance, secondiparous cows showed the lowest pre-calving body condition score in a study assessing 24,807 Holstein or Holstein Friesian cattle from three countries [46], thus potentially leading to easier calving. Cows of a higher parity have been shown to produce larger calves [47], and are more prone to the development of metabolic disorders [48,49], which may negatively affect calf viability or the calving process [45]. Further research is needed to further evaluate this observed trend, but it seems obvious that optimal heifer management, prevention of over-condition or metabolic diseases can serve in reducing perinatal calf mortality.

## 4.6 Farm size

A limited number of studies have investigated the potential association between farm size and perinatal mortality (PM), and the results are hardly comparable, mostly because of large variability in herd size or PM definitions and the differences in applied statistical methods. Most studies that investigated farm size as a predictor of perinatal mortality (PM) on the individual animal level categorized the size into discrete groups, as analyzing it numerically on the animal level is not meaningful. However, a review by Cuttance and Laven [9] identified some studies that analyzed farm size as a numerical variable on the farm level, and none of these identified a significant association with PM. The inconsistent categorization of farm size, the lack of a significant association between farm size and PM on the farm level in several studies [6,13,15],

and the limited number of farms analyzed have made this predictor less informative compared to other previously described predictors.

On the individual animal level, some authors [6] did not observe a significant relationship between PM and herd size categories, while others [13] observed that farm size was significant in a univariable analysis but no longer showed significance after inclusion in a multivariable model. While we observed a significant relationship between farm size and PM on the cow level in our multivariable analysis (*P value* = 0.043) (with an increased risk of perinatal calf mortality with increasing farm size), the level of significance was borderline and thus needs to be interpreted carefully. Gulliksen and co-workers [15] also found that larger farms had higher odds of perinatal mortality (PM) compared to smaller farms in Norway. These findings align with our own results despite differences in definitions of the various size categories.

In the current literature there is a lack of consensus regarding the definition of small and large farms. The classification of farm size is often highly subjective and can vary based on the specific dataset and the researcher's personal interpretation, or the typical farming structure in the area examined. Previous studies have used arbitrary farm size categories, which may not provide meaningful results. To address this issue, we employed a statistical binning technique. While the binning technique aims to mitigate personal bias, it is still subjective and may vary based on the specific dataset. Therefore, it is essential to establish a clear and meaningful definition of farm size, as farm size appears to be a significant predictor for PM, particularly when interactions are considered, such as between farm size and calving ease as observed in our study. It is also important to identify associated management or husbandry factors which are related to a higher PM risk on larger farms in order to reduce that risk. While larger farms are more likely to enjoy the benefits of better technology, such as, for instance, camera surveillance, they may suffer from poorer staff commitment in comparison to family-run, smaller farms where the farmer is likely to be more committed to a positive calving outcome than an employee. Since we could not assess data on available technology, work force or calving management on the studied farms, this however remains speculative. Evaluation of these and other farm-specific traits or management approaches as potential risk factors for PM, particularly on large farms, is thus highly desirable to identify and subsequently address these issues.

## 4.7 Limitations of the study

This study involved the retrospective analysis of a pre-existing dataset which was created in the context of a different study [21]. Prospective planning of on-farm data recording and data acquisition within the specific context of perinatal mortality was therefore not possible. The study also had to rely on farmer-recorded data. In Germany, farmers are obliged to record the birth of a full-term calf within seven days of birth. Entry is required as either stillborn or live-born, and later calf deaths also need to be recorded. The registration of abortions is not required, data on abortions were therefore not available. There is naturally some uncertainty as to the accuracy of farmer-recorded data, particularly with regard to distinguishing between very late abortions and stillbirth at term, with regard to the accuracy of exactly fulfilling the definition of postnatal mortality "within 48 hours" as part of PM or with regard to consistent classification of parameters such as "calving ease" by the different farmers involved. Large-scale studies however cannot be performed without the use of farmer-recorded data and the use of large, relatively anonymous datasets, and many previous studies have relied on similar regional or national databases [10,11,33–35]. Similar limitations therefore also apply to these previous studies. Another limitation applying to the raw data was the restriction of the available data set to a 12 month period per farm. Potential effects of the individual cow on PM risk could therefore not be assessed, as the vast majority of the animals only contributed one

calving record to the database. In addition, corresponding insemination dates were largely unavailable, precluding calculation of gestation length. Calving records were also only available on the parturition level, leading to inadequate records for individual calves born as twins or multiples. Previous research has indicated that twins and triplets have a higher risk of PM compared to singletons when accurately recorded [15]. Due to the inadequate recording of these variables our study results are limited to singleton calves. The evaluation of crucial variables such as gestation length and twinning on PM was thus not possible from the given data set.

Birth weight, another interesting potential predictor, was not recorded, and calf sex, while part of the database, showed a substantial proportion (84.5%) of missing values amongst still-born calves. Farmers often see stillbirth or early calf mortality as inevitable and something that cannot be changed, so the feeling that details of the perished calf (such as calf sex) do not really matter will lead to less attention being paid to data recording, highlighting a probable reason for the under-examination of topics such as stillbirth and perinatal mortality. An attempt at imputing missing values for calf sex proved to be challenging, leading to the exclusion of this parameter from the analyses. This is unfortunate, as the impact of calf sex on PM remains inconclusive from previous studies, with 40% of studies reporting an effect and 60% reporting no effect [5].

Despite these limitations, the analysis revealed important information on risk factors of perinatal mortality in dairy cattle in Germany while simultaneously assessing different statistical methods for the analysis of large data sets.

## 5 Conclusions

The most crucial predictor for PM, identified by all modelling approaches, was calving ease, with difficult calving leading to a significantly higher perinatal mortality risk. Two significant interactions indicated that the effect of calving ease was however influenced by breed and farm size. The Simmental breed showed the lowest PM risk compared to other breeds for difficult and medium difficult parturitions. Surprisingly, larger farms only experienced a higher PM risk compared to smaller ones when calving was difficult; for easy or medium calving, farm size did not play a significant role. These key results highlight the importance of complex statistical methods to fully understand clinical data sets. We recommend using mixed-effects logistic regression for analyzing perinatal mortality for multi-farm or multi-heard data due to its ability to account for farm-level variations. Although random forest can be helpful, it requires careful data stratification. For robustness and an increased level of inference, we advocate brute-force variable selection over backward/forward selection. In summary, our study sheds light on some of the intricate factors influencing perinatal mortality in calves. Data limitations however prevented the analysis of further crucial factors such as birth weight, calf sex and twinning, and further studies are warranted to study these predictors. We therefore emphasize the need for ongoing research of perinatal mortality on the animal level, and for uncovering risk contributors to perinatal mortality on the farm level.

## Supporting information

**S1 File. Supplementary materials.**
(DOCX)

## Acknowledgments

We would like to thank Andreas Oehm and Gabriela Knubben-Schweizer for helpful discussions.

## Author Contributions

**Conceptualization:** Yury Zablotski, Martina Hoedemaker, Annegret Stock.

**Data curation:** Yury Zablotski, Katja Voigt, Laura Kellermann, Heidi Arndt, Maria Volkmann, Linda Dachrodt, Annegret Stock.

**Formal analysis:** Yury Zablotski, Katja Voigt.

**Funding acquisition:** Martina Hoedemaker.

**Methodology:** Yury Zablotski, Kerstin E. Müller.

**Project administration:** Martina Hoedemaker.

**Resources:** Martina Hoedemaker, Kerstin E. Müller.

**Software:** Yury Zablotski.

**Supervision:** Yury Zablotski, Katja Voigt, Martina Hoedemaker, Annegret Stock.

**Validation:** Yury Zablotski, Katja Voigt.

**Visualization:** Yury Zablotski.

**Writing – original draft:** Yury Zablotski, Katja Voigt.

**Writing – review & editing:** Yury Zablotski, Katja Voigt, Martina Hoedemaker, Kerstin E. Müller, Laura Kellermann, Heidi Arndt, Maria Volkmann, Linda Dachrodt, Annegret Stock.

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
