## [Decision Letter · Decision Letter 0]

18 Jan 2024

PONE-D-23-30463Risk factors influencing perinatal mortality in dairy cattle in Germany, validated by modern modelling approaches: uni- and multivariable mixed-effects logistic regressions, random forest and brute-force variable selection methodsPLOS ONE

Dear Dr. Zablotski,

Thank you for submitting your manuscript to PLOS ONE. After careful consideration, we feel that it has merit but does not fully meet PLOS ONE’s publication criteria as it currently stands. Therefore, we invite you to submit a revised version of the manuscript that addresses the points raised during the review process. Three experts have reviewed your manuscript (please note 2 reviewers have submitted their detailed comments on attached documents). They all have raised relevant concerns for the manuscript to be considered further. I concur with their view. All the reviewers' comments would need to be addressed in your revision. Overall, I believe addressing the reviewers' comments will enhance the quality of the manuscript.

We look forward to receiving your revised manuscript.

Kind regards,

Angel Abuelo, DVM, MRes, MSc, PhD, DABVP (Dairy), DECBHM

Academic Editor

PLOS ONE

Journal Requirements:

2. You indicated that ethical approval was not necessary for your study. We understand that the framework for ethical oversight requirements for studies of this type may differ depending on the setting and we would appreciate some further clarification regarding your research. Could you please provide further details on why your study is exempt from the need for approval and confirmation from your institutional review board or research ethics committee (e.g., in the form of a letter or email correspondence) that ethics review was not necessary for this study? Please include a copy of the correspondence as an ""Other"" file.

6. We note that Figure S1 in your submission contain map/satellite image which may be copyrighted. All PLOS content is published under the Creative Commons Attribution License (CC BY 4.0), which means that the manuscript, images, and Supporting Information files will be freely available online, and any third party is permitted to access, download, copy, distribute, and use these materials in any way, even commercially, with proper attribution. For these reasons, we cannot publish previously copyrighted maps or satellite images created using proprietary data, such as Google software (Google Maps, Street View, and Earth). For more information, see our copyright guidelines: http://journals.plos.org/plosone/s/licenses-and-copyright.

a. You may seek permission from the original copyright holder of Figure S1 to publish the content specifically under the CC BY 4.0 license.  

Reviewers' comments:

Reviewer's Responses to Questions

**Comments to the Author**

1. Is the manuscript technically sound, and do the data support the conclusions?

Reviewer #1: Yes

Reviewer #2: Partly

Reviewer #3: Yes

2. Has the statistical analysis been performed appropriately and rigorously? 

Reviewer #1: Yes

Reviewer #2: I Don't Know

Reviewer #3: Yes

3. Have the authors made all data underlying the findings in their manuscript fully available?

Reviewer #1: Yes

Reviewer #2: Yes

Reviewer #3: No

4. Is the manuscript presented in an intelligible fashion and written in standard English?

Reviewer #1: Yes

Reviewer #2: Yes

Reviewer #3: Yes

5. Review Comments to the Author

Reviewer #1: This work about perinatal mortality represents valuable information on risk factors and a creative statistical methodology to evaluate perinatal mortality. The reviewer congratulates the authors for such as extensive work. This reviewer also considers that the manuscript needs a re-organization of the sections and a detail observation to correct grammatical mistakes. Please see the comments below:

L 30 - This definition is imcomplete, add "proportion of" since you are reporting a percentage.

L 39: Ideally a key word should not be in the title of the manuscript.Change it, please

L 43: why? could you further explain why this is still a problem? has not been enough research done? what about values of perinatal mortality worldwide, haven't those changed in the past years? add more information please

L 44 - 46: Please, each of these statements should be accompanied by a citation. Are there necessary losses of animals? maybe rephrase this sentence. Why PM raises concern on animal welfare? How can you measure that? Please explain.

L 47- 49: I think this idea should separate from the above stated in L 47. Do you mean studies are difficult to compare? If I run a study with my own definition of PM I can still asses risk factors, however It'll be complicated to compare with other studies. Please rephrase this section.

L 47: Do you mean associated factors? if you know factors that causes PM, please mention them briefly, also, indicate if these are at the calf level or at herd level

L 50: please, give examples for this.

L 49: do you mean a stillbirth?

L 51: definiton, not a figure. Please change

L 58: if there is a percentage, there is a denominator, what is the denominator for this study? stillbirth, etc over what?

L 60 - 61: Avoid the word recent, in 10 years that study won't be recent. Also, how many studies were analyzed by Cuttance, was this a literature review, systematic review, metananalysis, this information is important please add.

L 62: Not sure if that is beacuse of just different definitions, don't you think that dairy industry practices changes regionally? that could cause that risk factors in Europe for PM are not the same that in SouthAmerica for example. Add this in your introduction please.

L 66: Your introduction bullet points are everywhere, please re organize your ideas for the introduction, this should be added above when you mention the study.

L 73: What is a classic Logistic regression? are you saying using a general model approach instead of mixed model approach? Avoid writing vague terms, below you explain mixed models, but you did not explain what is a classic logistic regression.

L 75- 79: Here I can argue your statement, logistic regression assess the probability of a binary outcome to happen, and you are measuring a continuos outcome, PM, what would you use logistic regression in the first place? What about linear regression, Poisson regression methods...Just mention that some statistical methods are questionable,or rephrase this paragraph.

L 87: This is interesting, I wonder if you add information about whats the percentage of farms that use these breeds in Germany, so you add more relevance to your study.

L 100: this should be in your methods, not here

L 102: what does wider context mean? Maybe just refered to Describe PM?

L 107 - 120: This should be in the methods and not here. Normally your objectives should be in the end of your introduction. Also, where is the hypothesis for the study?

L 132: what was your sample size based on?

L 134: Could you explain in further detail how this farms were selected? Who made the randomization? What where the relevant organization?

L 140: I am wondering if a human ethics guideline was done previous the survey, if so, please, could you add that to the manuscript.

L 143: How many researchers? were all of them trained to do the survey? were all the surveys done in person?

L 155: similar comment as above, rephrase the definition please.

L 167: Do you mean that the predictor "calving ease" was accepted to be included in the analysis?

174: I agree with the logic of your decision to exclude these variables, however in your definition of PM you included stillbirth, without not knowing the calving date, how did you know which cases were stillbirths? was this recorded by the farmer, if so, be clear to state that the stillbirth definition was based on the farmer's criteria.

L 185: Thank you for reporting the median and the IQR. Herd size definition is missing so far, please add. Did you have a herd with only one cow? Also, why did you choose to use binned quantiles and not a 25% quantile? could you briefly explain? the same with other parts of your methodology, you are introducing a creative way to analyze risk factors associated with PM, thus I would expect a more detail explanation.

L 196: extra parentheses

L 204: So far, I did not understand what type of level was being analyzed, it seems that individual level data was the approach use, could you please refer to this above when describing the data.

L 239: I understand your statistical approach, however it is confusing how it is described. I suggest to start by explaining you dependent variable, then explain the collinearity, univariable models, final models, testing for confounders and interactions and multiple comparisons. Thereafter explain the AIC and BIC and the Random Forest algorithm. Or rephrase the statistical methods for a better understanding.

L 266: This adds to a 99%, please complete

L 269: Maybe rephrase so all the percentages are in number format.

L 285: If season was not significant in the univariable model, why did you force it into the final model? could you please explain?

Figure 4.1: Please add more information in the title, for example, number of farms, animals,.... be more descriptive. Could you add information (labels) about what the values of the x axis represent, they are different between the graphs.

Figure 4.2: All figures need more description in in their captions.

L 416: What about compared to PM values measured before in Germany or other countries?

L 415: According to your definition, what you measure was risk of PM, please use that measure when mentioning your PM result.

L 430: Good sentence!

L 433: How? could you compare a study that define PM within the first 24 h of life compared to other study that defined PM within the first 48 of life? be specific about your method can be useful for comparisons.

L 442: A suggestion, maybe remove "others" if the purpose of this risk factor was to assess diferences between different breeds and PM, others is not useful.

L 447: Please add more information about number of animals in each study for example: 9.5% (150/975).Also, are all this studies on calf-level data? just to have fair comparisons...

L 452: this is very important for your work as one of your objectives is to use another analysis approach for PM. Maybe mention what type of analysis those studies used (not all of them but some) and then put it together with what you have written from L 452 to 458.

L 451: Again, use risk instead or rates as you did not calculate rate of PM or at least that was not stated in the manuscript.

L 461: This is a big statement to make with such low number of Simmental animals in your study. Even later in L 465 you mentioned that is not known why, therefore I would say that your data suggests... and stick with what you mention in L 467 – 472

L 472: Don't start with an abbreviation.

480: This sounds unlikely to be performed, think about it, you need different farms with the same management, or a large dairy farm with Simmental which is unlikely as well. Maybe mention the importance of observational studies for this purpose.

L 488: I would delete this sentence.

L 490 - 500: Here there is something missing, what criteria did the other studies use to determine season, was it the same as the one used in this manuscript

L 503 - 511: Did all of those studies use logistic regression? is logistic regression the only way to analyze perinatal mortality?

L 512 - 516: What about farm management practices or facilities, are they the same between the U.S and Norway or Germany?

L 574: increased when, at the second parity? add the information.

L 581: There are different terms for parity, stick with one of them please

L 592: please rephrase.

L 606: is this odds ratio, risk ratio? just mention the "odds" or the "risk". Also, I would suggest to write the 95% CI like this: 95% CI = 1.08 - 1.43.

L 615: Why?

L 627: I found this a valid reason for the differences of the effect of farm size and PM, in some countries larger farms have more technology for example. Would not you think that also has an impact?

L 631: please add this above in the description of your results as well.

L 642: are you suggesting to add farm as a random effect when analyzing herd level data? be more clear in this sentence.

L 695: Please stick only with mentioning the limitations of the study and not the solutions for it, the paper its already long and this could help to make it shorter.

The reviewer considers that this manuscript can be shorter as some parts are long and repetitive, consider this when answering the comments above.

Reviewer #2: Thank your for your manuscript. I think there is just some more detailed information missing to better understand what you did. I do not have the impression that the study itself is not sound. Happy to receive the revised version.

Reviewer #3: Review is available as an attachment

6. PLOS authors have the option to publish the peer review history of their article (what does this mean?). If published, this will include your full peer review and any attached files.

Reviewer #1: No

Reviewer #2: No

Reviewer #3: No

---

## [Author Response · Author response to Decision Letter 0]

29 Feb 2024

Reply to editor comments:

Thank you for submitting your manuscript to PLOS ONE. After careful consideration, we feel that it has merit but does not fully meet PLOS ONE’s publication criteria as it currently stands. Therefore, we invite you to submit a revised version of the manuscript that addresses the points raised during the review process.

Thank you for your encouraging assessment. We are grateful to the three reviewers for their constructive review and feel their comments have helped us to significantly improve the manuscript. We have addressed all comments in the revised version of the manuscript and have replied to all editor and reviewer comments in a point by point fashion below.

2. You indicated that ethical approval was not necessary for your study. We understand that the framework for ethical oversight requirements for studies of this type may differ depending on the setting and we would appreciate some further clarification regarding your research. Could you please provide further details on why your study is exempt from the need for approval and confirmation from your institutional review board or research ethics committee (e.g., in the form of a letter or email correspondence) that ethics review was not necessary for this study? Please include a copy of the correspondence as an ""Other"" file.

A written Ethical Statement is provided by the leading scientist Prof. Hoedemaker, who was responsible for the study as an “Other” file.

We included a statement in the Methods section.

The data are available upon request from the corresponding author.

We included a statement into the Methods section.

6. We note that Figure S1 in your submission contain map/satellite image which may be copyrighted. All PLOS content is published under the Creative Commons Attribution License (CC BY 4.0), which means that the manuscript, images, and Supporting Information files will be freely available online, and any third party is permitted to access, download, copy, distribute, and use these materials in any way, even commercially, with proper attribution. For these reasons, we cannot publish previously copyrighted maps or satellite images created using proprietary data, such as Google software (Google Maps, Street View, and Earth). For more information, see our copyright guidelines: http://journals.plos.org/plosone/s/licenses-and-copyright.

 ].”

Since the map did not contain essential information, we have opted to remove it from the manuscript. 

Below we add replies to all three reviewers:

Reviewer #1: 

This work about perinatal mortality represents valuable information on risk factors and a creative statistical methodology to evaluate perinatal mortality. The reviewer congratulates the authors for such as extensive work. This reviewer also considers that the manuscript needs a re-organization of the sections and a detail observation to correct grammatical mistakes. Please see the comments below:

Thank you very much for your encouraging review and for your valuable time and suggestions. We addressed all your comments to the best of our ability and feel they have definitely improved our manuscript. All changes are outlined in detail below and highlighted in the revised manuscript. Please note: any line numbers refer to the original submission, since line numbers have changed in the revised version of the manuscript.

L 30 - This definition is imcomplete, add "proportion of" since you are reporting a percentage.

Added.

L 39: Ideally a key word should not be in the title of the manuscript.Change it, please

The key words have been revised to ensure they no longer include any terms used in the title. In addition, due to a request by another reviewer to make the title “more compact, more attractive and sparking more interest” the title has been changed to: “Perinatal mortality in German dairy cattle: unveiling the importance of cow-level risk factors and their interactions using a multifaceted modelling approach.” 

L 43: why? could you further explain why this is still a problem? has not been enough research done? what about values of perinatal mortality worldwide, haven't those changed in the past years? add more information please

More information has been added to this section: this is still a problem despite a high research interest, and the incidence risk for PM has increased rather than decreased over time as shown by a systematic review and meta-analysis of the subject.

L 44 - 46: Please, each of these statements should be accompanied by a citation. Are there necessary losses of animals? maybe rephrase this sentence. Why PM raises concern on animal welfare? How can you measure that? Please explain.

Citations have been added to support these statements and further information has been added to explain why there are concerns about animal welfare regarding PM

L 47- 49: I think this idea should separate from the above stated in L 47. Do you mean studies are difficult to compare? If I run a study with my own definition of PM I can still asses risk factors, however It'll be complicated to compare with other studies. Please rephrase this section.

This section has been re-phrased to clarify this.

L 47: Do you mean associated factors? if you know factors that causes PM, please mention them briefly, also, indicate if these are at the calf level or at herd level

We are sorry for the confusion. We will talk about risk factors later. Here we just wanted to introduce the topic of definitions. So, we removed “This is caused, amongst other factors, …” and changed it to “A major difficulty for the comparability of studies is caused…”. We hope this clarifies the issue.

L 50: please, give examples for this.

References to individual studies have been added to give examples of these varying definitions. 

L 49: do you mean a stillbirth?

Yes, by “death of a full-term calf during parturition” we mean stillbirth. The term has now been added in brackets to clarify. The definitions of perinatal mortality used in previous publications often (but not always) cover stillbirths plus post-natal deaths to include a varying post-natal time frame. 

L 51: definition, not a figure. Please change

Thanks! Done.

L 58: if there is a percentage, there is a denominator, what is the denominator for this study? stillbirth, etc over what?

The denominator is the total number of still- and live born calves. The sentence has been adjusted to clarify.

L 60 - 61: Avoid the word recent, in 10 years that study won't be recent. Also, how many studies were analyzed by Cuttance, was this a literature review, systematic review, metananalysis, this information is important please add.

We deleted “recent”, added “26” before “different studies” and replaced “review of the literature” by “literature review”.

L 62: Not sure if that is beacuse of just different definitions, don't you think that dairy industry practices changes regionally? that could cause that risk factors in Europe for PM are not the same that in SouthAmerica for example. Add this in your introduction please.

Yes, we also think that regions could differ. Reference to variation in farm characteristics has been added here and had also already been referred to in line 69 of the original manuscript where we wrote that “… the challenges in comparing and interpreting the results due to the high variability in methods of data collection, definitions of PM [1,2], farm characteristics and locations, inconsistent inclusion of PM predictors, and the use of highly variable or simple statistical methods …”. 

The risk factors may differ in different locations even when the definition is the same. But we will only be able to compare locations properly when we have the same definition. 

L 66: Your introduction bullet points are everywhere, please re organize your ideas for the introduction, this should be added above when you mention the study.

We attempted to re-organize and re-phrase parts of the introduction to create a better and more consistent line of thought.

L 73: What is a classic Logistic regression? are you saying using a general model approach instead of mixed model approach? Avoid writing vague terms, below you explain mixed models, but you did not explain what is a classic logistic regression.

This has been re-worded to clarify that we mean logistic regression without the inclusion of random effect. 

L 75- 79: Here I can argue your statement, logistic regression assess the probability of a binary outcome to happen, and you are measuring a continuos outcome, PM, what would you use logistic regression in the first place? What about linear regression, Poisson regression methods...Just mention that some statistical methods are questionable,or rephrase this paragraph.

This seems to be a misunderstanding. PM is binary, 0 or 1, alive or dead. Not continuous. The Mixed-Effects logistic regression also assesses the probability of a binary outcome. The only, but decisive, difference between logistic and mixed-effects logistic regression is the random effect on individual farms. 

L 87: This is interesting, I wonder if you add information about whats the percentage of farms that use these breeds in Germany, so you add more relevance to your study.

We have added some information about the frequency of these breeds to the introduction. 

L 100: this should be in your methods, not here

You are right, thank you! Since this information was already part of the M&M, the duplication has merely been removed from the introduction. 

L 102: what does wider context mean? Maybe just refered to Describe PM?

the term “in a wider context” has been removed from the sentence. 

L 107 - 120: This should be in the methods and not here. Normally your objectives should be in the end of your introduction. Also, where is the hypothesis for the study?

You are right. We moved this part to the end of statistical part of M&M, reduced it to avoid repetitions and added the hypothesis at the end of introduction.

L 132: what was your sample size based on?

Details regarding the calculation of the sample size have been added to the M&M.

L 134: Could you explain in further detail how this farms were selected? Who made the randomization? What where the relevant organization?

More detail on farm selection has been added. 

L 140: I am wondering if a human ethics guideline was done previous the survey, if so, please, could you add that to the manuscript.

We added a statement that no approval by an animal or human ethics committee was required for this study.

L 143: How many researchers? were all of them trained to do the survey? were all the surveys done in person?

The number of researchers has been added. Reference to training of the researchers has been added. The text already explains that all farms were visited in person.

L 155: similar comment as above, rephrase the definition please.

The denominator is the total number of still- and live-born calves. The definition has been re-phrased to clarify.

L 167: Do you mean that the predictor "calving ease" was accepted to be included in the analysis?

We mean than imputation for missing values for calving ease was accepted. The sentence has been clarified accordingly.

174: I agree with the logic of your decision to exclude these variables, however in your definition of PM you included stillbirth, without not knowing the calving date, how did you know which cases were stillbirths? was this recorded by the farmer, if so, be clear to state that the stillbirth definition was based on the farmer's criteria.

Sorry for the confusion. Calving date was known for all of the data, and we have added this to the list of available data for analysis. We also added a statement to the M&M section that the diagnosis of stillbirth was based on the farmer’s assessment to clarify to the reader that the researchers had to rely on farmer-recorded data. In addition, discussion of the topic of farmer-recorded data has been added

L 185: Thank you for reporting the median and the IQR. Herd size definition is missing so far, please add. Did you have a herd with only one cow? Also, why did you choose to use binned quantiles and not a 25% quantile? could you briefly explain? the same with other parts of your methodology, you are introducing a creative way to analyze risk factors associated with PM, thus I would expect a more detail explanation.

Some explanation has been added to the text to explain that there are no universally accepted farm size categories, and why the binning technique was chosen. Splitting into 3 categories was done by the quantile-method, which assures that all three categories have approximately equal numbers of observations. In our case the quantile binning split our dataset into 255 (large), 253 (medium) and 257 (small) farms, which is 3 groups of approx. 33% each. We did not want to categorize the farms arbitrarily, that is why we used the statistical technique – this is now explained in the text. We think that 3 farm size categories (small, medium & large) are more intuitive than 4 categories, that is why we did not use a 25% quantile.

Yes, you are right: a well accepted definition of farm size is missing in the literature so far. That is what we discuss later in the manuscript. For our dataset, the herd size definitions are given in lines 187-188 of the original text. 

Yes, there were three farms with only 1 dairy animal. Another farm had 3 cows. The rest of the farms in the small-farm-category (1 – 52 animals) had 4+ cows. 

L 196: extra parentheses

We re-phrased this to remove the extra parentheses 

L 204: So far, I did not understand what type of level was being analyzed, it seems that individual level data was the approach use, could you please refer to this above when describing the data.

You are right, individual level data was analyzed. Therefore, we added this information in several places before Line 204, including the title and abstract.

L 239: I understand your statistical approach, however it is confusing how it is described. I suggest to start by explaining you dependent variable, then explain the collinearity, univariable models, final models, testing for confounders and interactions and multiple comparisons. Thereafter explain the AIC and BIC and the Random Forest algorithm. Or rephrase the statistical methods for a better understanding.

Thank you for this suggestion. We now state PM as a single response variable. We also say that there were initially more variables considered. Then describe that breed and region are collinear. And finally state that 5 variables remained in the model as non-collinear according to VIF. We only then come to modelling and random forest + brute-force at the end. We hope this addresses your concern!

L 266: This adds to a 99%, please complete

Thank you for highlighting this! We now use the decimals and the percentages add up to 100%.

L 269: Maybe rephrase so all the percentages are in number format.

This section has been re-phrased accordingly 

L 285: If season was not significant in the univariable model, why did you force it into the final model? could you please explain?

There are two reasons. First, we wanted to ensure the comparability of 4 different methods regarding how they all evaluate all predictors and how important they “think” each predictor is. Secondly, after comparing two multivariable models, one with all 5 predictors and another with only 4 without season (see below), they did not differ significantly (p = 0.4), so that keeping season in the multivariable model did not significantly reduce the quality of the model. 

Figure 4.1: Please add more information in the title, for example, number of farms, animals,.... be more descriptive. Could you add information (labels) about what the values of the x axis represent, they are different between the graphs.

The caption has been made more informative and the suggested information has been added. Particularly, we added the following text: “All analyses were based on 133,942 parturitions from 721 farms.” 

By x-values you probably mean units? If so, the Chi-Square Statistic from a model usually takes the units of the response variable. For example, if our dependent variable is milk yield measured in kilograms, then the units of Sum Sq (this is what Chi-Square Statistic is) would be kilograms squared (kg^2). But, since we have mixed-effects logistic regression, our response is calculated in log-odds, and they are unitless. 

Figure 4.2: All figures need more description in in their captions.

We added more description to all figure captions. 

L 416: What about compared to PM values measured before in Germany or other countries?

We have added a brief comparison to previous German studies. 

L 415: According to your definition, what you measure was risk of PM, please use that measure when mentioning your PM result.

We have changed all “PM rates” to “PM risk” throughout the paper, wherever model results are reported or referred to.

L 430: Good sentence!

Thank you! 

L 433: How? could you compare a study that define PM within the first 24 h of life compared to other study that defined PM within the first 48 of life? be specific about your method can be useful for comparisons.

This sentence meant to say that use of several statistical methods would increase the comparability of our study to other studies. And if more studies would follow our example, we would have better comparability in the future. However, the discussion has been re-arranged, partly shortened and changed in focus at the request of another reviewer, so this sentence has been deleted in this process. 

L 442: A suggestion, maybe remove "others" if the purpose of this risk factor was to assess diferences between different breeds and PM, others is not useful.

We agree there was too much focus on “others” in this section of the discussion. Since the discussion needed to be shortened, the relevant section has been removed in the process. 

L 447: Please add more information about number of animals in each study for example: 9.5% (150/975).Also, are all this studies on calf-level data? just to have fair comparisons...

The numbers have been added and two more countries have been added for low mortality. 

Yes, all of them are animal-level data. This information has been added.

L 452: this is very important for your work as one of your objectives is to use another analysis approach for PM. Maybe mention what type of analysis those studies used (not all of them but some) and then put it together with what you have written from L 452 to 458.

We also agree that discussion of the methodical part is very important, so that we singled out the whole chapter in the discussion “Modelling framework”. And we think adding the comparison to other studies fits there the best. Following changes to the discussion, this section about modelling framework has been moved to the top of this section. 

L 451: Again, use risk instead or rates as you did not calculate rate of PM or at least that was not stated in the manuscript.

We changed all “PM rates” to “PM risk” throughout the paper, wherever model results are reported or referred to.

L 461: This is a big statement to make with such low number of Simmental animals in your study. Even later in L 465 you mentioned that is not known why, therefore I would say that your data suggests... and stick with what you mention in L 467 – 472

We weakened the statement by using “suggest a potential” instead of “highlights the potential” and rephrased the sentence. 

L 472: Don't start with an abbreviation.

SIM was changed to “Simmental”.

480: This sounds unlikely to be performed, think about it, you need different farms with the same management, or a large dairy farm with Simmental which is unlikely as well. Maybe mention the importance of observational studies for this purpose.

Following re-arrangement and re-wording of the discussion, this sentence has been deleted

L 488: I would delete this sentence.

Deleted.

L 490 - 500: Here there is something missing, what criteria did the other studies use to determine season, was it the same as the one used in this manuscript

The geographical description of the discussed papers was missing. So, we added a few more sentences to explain that the vast majority of studies involving season were conducted in the Northern hemisphere with similar seasonal variation. We hope we could address your comment. 

L 503 - 511: Did all of those studies use logistic regression? is logistic regression the only way to analyze perinatal mortality?

Almost all of them did. In the following lines (512-516 of the original manuscript) we write that only 2 studies used the mixed-effects logistic regression. 

The binary outcome limits the choice of models to logistic regression and mixed-effects logistic regression. Thus, yes, the models with only binomial family are the only way to analyze a binary outcome, such as PM. 

L 512 - 516: What about farm management practices or facilities, are they the same between the U.S and Norway or Germany?

That is an important question to ask but difficult to address. In general, we would deem those management practices largely comparable, but of course, management practices differ across countries. They however also differ widely across farms in the same country. And that is why we used (and promote the use of) individual farms as the random effect. We feel that discussing this topic of cross-country comparability of husbandry and management would unnecessarily lengthen the paper, which is already very long, and have therefore not included discussion of this topic.

L 574: increased when, at the second parity? add the information.

L 581: There are different terms for parity, stick with one of them please

To address both comments above, we changed the sentence in the following way: “Interestingly, our model predicted a pattern similar to that found by Bicalho et al., where the probability of PM declined in secondiparous cows as compared to primiparous, but then increased again in multiparous animals.” We also changed the terms for parity to primiparous, secondiparous and multiparous in all sections of the paper to be consistent in the wording.

L 592: please rephrase.

This sentence has been deleted following re-structuring and shortening of the discussion.

L 606: is this odds ratio, risk ratio? just mention the "odds" or the "risk". Also, I would suggest to write the 95% CI like this: 95% CI = 1.08 - 1.43.

This detail has been deleted at the request of another reviewer to shorten the discussion and to remove repetition of results and other unnecessary detail

L 615: Why?

Because farm-sizes do not produce a nice distribution. There are usually either big or small farms. Numerical farm-size would also produce patches of several similarly sized farms, but not a normal pattern. Besides, a potential association would probably not be linear. Having 3 categories would uncover a non-linear pattern if one exists, or also show a linear pattern, if there is a steady change from the small to the big farms.

L 627: I found this a valid reason for the differences of the effect of farm size and PM, in some countries larger farms have more technology for example. Would not you think that also has an impact?

We agree that technology, workforce and personal commitment of farmers or staff (to name just a few factors) are likely to have an impact on PM. While larger farms are more likely to enjoy the benefits of better technology (this is also the case in Germany), they may suffer from poorer staff commitment in comparison to family-run, smaller farms where the farmer is likely to be more committed to a positive calving outcome than an employee. Since we do not have any data on available technology or work force on the studied farms, this however remains speculative. Since larger farms had a higher risk of PM in our study, the potential benefits of better technology do not seem to mitigate other drawbacks on larger farms. We personally think that farmer/staff commitment is key to reducing the risk of PM and has most likely a larger impact than technology, – brief discussion of this topic has been added.

L 631: please add this above in the description of your results as well.

It is already part of M&M on lines 187-188 of the original manuscript. We think it is more a methodical part than a result and have therefore opted against repeating this information in the results section. 

L 642: are you suggesting to add farm as a random effect when analyzing herd level data? be more clear in this sentence.

This line does not refer to the analysis of herd-level data. We have clarified this by adding “on the cow level” to the introductory sentence. 

L 695: Please stick only with mentioning the limitations of the study and not the solutions for it, the paper its already long and this could help to make it shorter.

Thanks, that is a good idea. We have now deleted parts of the text.

The reviewer considers that this manuscript can be shorter as some parts are long and repetitive, consider this when answering the comments above.

We have tried to delete any unnecessary repetition in an attempt to shorten the text. Addition of further detail on some topics has however been suggested by other reviewers. Also, the discussion has been re-arranged, re-phrased and shortened and re-focused at the request of another reviewer.

Reviewer #2: 

This interesting paper is well written. Specific comments are found below mostly surrounding clarification.

Thank you very much! We greatly appreciate your time and effort and your valuable comments which helped us improve the manuscript! All changes are outlined in detail below and highlighted in the revised manuscript.

Title

Please, make the title more compact and attractive. In my opinion, it is not necessary to add the methods used (this is information for the Materials & Methods section). The title should spark the reader's interest.

Thank you for this suggestion! We hope this title is better: “Perinatal mortality in German dairy cattle: unveiling the importance of cow-level risk factors and their interactions using a multifaceted modelling approach.”

Abstract

You've outlined the two aims of the study. While you successfully discuss the first aim, the description of results and conclusions for the second aim, specifically the evaluation and comparison of different statistical methods, is currently missing. Could you please elaborate on this aspect with one or two additional sentences?

We apologize for the oversight and have now added a short section to the abstract. 

Introduction

At line 99, you give the number of calving records, but in my opinion, this is for the Material and Methods section.

Removed from introduction. 

Lines 107-110 should be relocated to precede the aims of the study, as they provide essential context. Additionally, lines 110-120 should be moved to the materials and methods section for better organizational coherence.

This makes sense, thank you. We moved this part to the end of the statistical part of M&M, then reduced it to avoid repetitions (in the M&M), and added the hypothesis at the end of the introduction (as requested by another reviewer) to round up the introduction. 

As part of this re-organization of the introduction, lines 107-110 have been deleted. 

Materials & methods

Line 180-181: Place figure captions in the manuscript text in read order, immediately following the paragraph where the figure is first cited. So, please, replace this caption to line 134.

Following comments of the editor, we removed the map (Fig. 3.1.) from the manuscript. There is therefore no longer a figure in the Material and Methods section.

Figure captions 4.1. and 4.2. had already been placed correctly, but the caption for Figure 4.3. has now been moved to one paragraph below. Now all figure legends are placed below the paragraph where they were first cited.

Line 136: provide the time period of the study please.

It was already there at lines 143-144 of the original submission. We have now added additional brief reference to the time period to the introductory sentence of this chapter. 

Line 138: Could you clarify the intended meaning of a response rate 'between 6 and 9%'? It would be helpful to provide the overall global response rate first and then specify the response rates for each involved region to offer a more detailed breakdown.

This sentence in the M&M part was actually a bit confusing, we apologize. It’s a result. Since we later also go into further detail in the results section, we have now removed the sentence in line 138 from the M&M section. 

Line 156: In the literature, perinatal mortality typically refers to stillbirths beyond day 260 of gestation and calves that died within 48 hours after birth. However, the text is not entirely clear on whether the included cases encompass only full-term stillbirths and deaths within 48 hours postpartum, or if it also includes stillbirths that occurred between 260 days and full term. Please provide clarification on this point.

An explanation has now been added to clarify that the classification as stillbirth (death during parturition at term) or post-natal death within 48 hours was based on the farmers’ assessment, and that exact data on gestational length or hour of death were not available from the data set. Abortions were not recorded in the studied databases and were not included in the analyses. Discussion on the issue and uncertainties of farmer-recorded data has also been added.

Line 156-157: Why were observed abortions not included in this definition? The current inclusion seems to result in a potential overestimation of the perinatal mortality rate. Please provide clarification on the rationale behind this decision?

We followed the standardized definition of PM proposed by Wong et al., which does exclude abortions and refers to the death of a calf either during parturition or within 48 hours of birth. 

In addition, abortions (the pre-term delivery of an immature calf) provide different information from perinatal mortality and are a different topic in our opinion. Also, the farmers were not required to record any abortions in the studied databases, information on abortions was therefore highly inconsistent and unavailable for many farms. 

The death of an aborted calf is not related to parturition management or neonatal management procedures. The abortion rate would provide information on underlying cow health issues or infectious diseases causing abortions, and is thus a separate topic. We have also added further information and discussion on the issue of dealing with farmer-recorded data.

Line 173: In the earlier section of the manuscript, the term 'number of calves per birth' was used. For consistency, please retain the same variable name throughout and replace 'litter size' with 'number of calves per birth.

Done. 

Line 174: please, provide information on how many multiple births were excluded.

We have added the requested information to this section. 

Line 175-176: Was crown-rump length measured to estimate the age of the fetus/calf at stillbirth? If so, could you please describe the methodology used for this measurement and provide relevant details? If not, could you provide an explanation for the omission of this measurement and the chosen alternative, if any?

The study relied on the retrospective analysis of farmer-recorded data, crown-rump length was thus not measured. A statement has been added to clarify this.

Line 182: why was the number of abortions not recorded? Please, explain this more in detail.

We made use of a pre-existing dataset that was initially collected for a different purpose and without the evaluation of perinatal mortality in mind. Also, this data set originated from databases generated by compulsory data recording. Farmers are not required to register any abortions towards these databases, that is why this information was not available. The databases only require the entry of the birth of a full term calf, and any calf mortality. We have added explanation and discussion of this matter to the M&M section and the discussion.

Line 186: Please provide the complete written form for abbreviations the first time they are used.

Done.

Line 193-195: I have reservations about the classification of calving ease. For instance, categorizing a caesarean section as difficult may not accurately reflect cases where the procedure's difficulty is potentially lower than instances involving multiple helpers and mechanical pulling tools. Could you clarify whether all caesarean sections are performed after attempting mechanical pulling or with assistance from several helpers? If not, should it be better to create a fourth category (calving after caesarean section)? It may be advisable to consider excluding cases involving fetotomy, as in most instances, mortality is not directly attributed to the fetotomy procedure itself. Typically, fetotomy is performed due to the death of a large calf that cannot be delivered manually. Would you agree?

We agree that foetotomy would typically be performed due to the death of a large calf that cannot be delivered manually. This situation would be considered dystocia (=difficult birth) due to foetal oversize, having resulted in the death of the calf as a consequence of this difficult parturition. We therefore think (and can say from an author’s own experience in performing foetotomies in cattle) that parturitions resulting in foetotomy can safely be considered of a high difficulty (otherwise the situation would have been solved by other means). As for caesarean section, this procedure will only be performed in practice if a calf cannot be delivered vaginally. Before a decision for surgical intervention is taken, manual delivery attempts have usually been undertaken and have failed. In the dairy breeds covered in this study, caesarean section will almost exclusively have been applied in such cases, following a difficult parturition that cannot be resolved by other means, and those means will have been exhausted prior to a decision for surgery. In practice, farmers will do a lot to avoid a caesarean section. Elective caesarean sections without any prior vaginal delivery attempts (such as customary in Belgium to deal with Belgian Blue cattle) are not common practice in Germany, and particularly not in dairy breeds. Although we do not have any additional information on any details of the force used in prior delivery attempts other than the farmer-recorded categories, we would also be confident to attribute any cases that result in caesarean section to a high difficulty category.

We agree with you that a difficult parturition eventually resolved by caesarean section may in some cases be less traumatic to everyone involved than a possibly excessive use of mechanical pulling tools (and have added discussion on this matter). But this limitation regarding a lack of details applies to all the assessments made by the farmers, and we can also not be 100% sure about the extent of force that was applied during parturitions assigned to the other categories, since the individual farmers’ perception of the levels of difficulty may vary (despite the definitions given to them to help them assign the cases). (Reference to the fact that calving records were farmer-recorded data has been added to the M&M section).

We therefore also agree with you that the categories are crude, but we have to rely on the farmers’ records and assessments. Within these records, parturition difficulty had in fact initially been recorded in four categories: easy, medium, difficult and, worst case: veterinary intervention by surgery (i.e. C section or foetotomy). This category was only recorded by the farmers as “surgery” and it thus combined caesarean sections and foetotomy, so we are unable to distinguish between these two possibilities within the surgical category. This surgical category also contained only 0.12% of all the cases. Since the “difficult” vaginal delivery category was also relatively small, since statistical evaluation of very small sub-groups is problematic, and since we are confident that situations resulting in either caesarean section or foetotomy can be called “difficult” calving, we made the decision to combine difficult vaginal deliveries with the surgical cases and to analyze them as a single “difficult parturition” category. We hope this clarifies the issue and you can follow our line of thought.

Line 208: Change 'p-value' to 'P value' (italicized with capital) throughout the entire manuscript. Please review the entire document for this correction.

Done.

Line 216: what do you mean with “explanatory” and “R2”? I suppose this is “conditional coefficient of determination R2” and “Marginal coefficient of determination” respectively? Please, make this more clear.

This section has been re-phrased and we clarified the section by adding “conditional R^2, marginal R2” into the brackets instead of just “R2”.

Line 219: please, mention how many were others in region South.

Done. 8% of GH and 3% of others in region South was added.

Line 223: please change > into ≥

Done.

Line 224-225. The information is partially available in lines 217-218. Please add details about others in the South at line 219.

We did. And thus deleted this sentence as repetition.

Line 228: Please make it easier to read by choosing either 'differences' or 'contrasts' and removing the other.

Since “difference” is more generic, we kept “contrasts” to describe differences between categories in categorical variables. 

Line 251-253: please, provide a reference for this.

The reference has been added here. 

Results

Line 294: in figure 4.1d, it looks like calving ease and parity have the same predictive value. Please provide the values in the text, to make this clearer.

The brute-force does not report predictive values. We did explain in the original paper in lines 251 – 253: “The importance value for a particular predictor or interaction is equal to the sum of the weights for the models in which the variable appears. Thus, a variable that appears in many models with large weights will receive a high importance value.” 

The statistical package only provides the plot, but not the values. And we think that the exact values are less important than the ranking of the variables. Two variables can then have the same importance, because they appear in most of the models and the weights of those models could sum up to the same number. Since we are not studied statisticians, but, as any scientist, need to use statistics for analysis, we can’t understand and explain the calculations to the deepest level. We would appreciate your understanding!

Line 335: if significantly, please provide P value.

We have now provided the P values.

Line 373: please provide P value

The P values are presented in table 4.4. Specific reference to this table has been added at the end of this sentence. As far as we know, repeating P values in the text when they are already part of the table is not recommended. Besides, there are usually over 10 contrasts for any interactions, so providing over 10 P values within the text would hinder readability. We therefore refer the reader to the table(s) for these details.

Line 374-375: please, rephrase this sentence, and provide the same information for each group.

Again, giving all P values within the text would significantly hinder readability, and writing one sentence for each contrast would make this section impossibly long, while other reviewers already recommended to shorten the paper. Moreover, these results are already presented in detail in two tables (4.3 and 4.4). We have now added two clear references to these tables to guide the reader to these results and have refrained from repeating the full contents of the tables in the text. 

Line 376-378: is this significant? If so, please provide P values.

See above – we are now clearly guiding the reader to the relevant tables, where this information is provided in detail. Due to the large number of contrasts, providing the P values for each contrast in the text is not possible without undue repetition and loss of readability. The text therefore introduces the tables, and the requested information is presented in the tables. 

Line 378: do you mean rate or incidence or prevalence? Please check this.

Thank you for highlighting this. We changed it to “risk of PM”

Line 381: please provide P value

Done. 

Line 387-391: are these results statistically significant? If not, you can not say “higher” or “lower”, but you should say “is numerically lower, although not significantly”. If it is significant, please provide P values.

You are right, thank you. The P values have been added. Please note, that through correction for multiple comparisons, some P values might become identical. This happened in this case, where all 3 P values round up to 0.021. But since the odds ratios differ, they are different tests and there is no mistake.

Line 393-404: provide P values were necessary (when results were statistically significant).

Similarly to the comments above, we think that listing all 9 P values for all 9 significant contrasts between all the categories inside this interaction in the text would greatly hinder readability and be an undue repetition of results. We have now clarified the previously available reference to table 4.4 by adding “Detailed results and P values are given in Table 4.4”. 

Discussion

In my opinion, it is not necessary to divide the discussion into separated subsections.

Thank you for this suggestion, which we considered carefully. Since the discussion is very long and multi-faceted, we however think that these subsections provide better clarity for the reader and can help guide them through this section. The other two reviewers did not object, so we have opted to keep these subsections unless advised otherwise by the editors. Also, the discussion has been revised, re-arranged re-focused and shortened at the request by another reviewer.

Line 440: I find it somewhat confusing that you state your study aligns with the findings of Voljc et al., but at line 442, you mention that the results for Brown Swiss deviate from this trend. Could you please provide clarification to make this aspect clearer? 

We agree, this section was indeed confusing. We have now re-worded this section to clarify. 

In each subsection, begin by presenting your main findings. Concentrate on highlighting your results first, and subsequently, engage in a discussion comparing your findings with those of relevant studies.

This suggestion unfortunately contradicts the suggestions by another reviewer, who asked us to remove any detailed reference to or repetition of results from the discussion and to add more interpretation within the veterinary context rather than comparison of results to other studies. We have thus re-worded and reviewed the discussion accordingly. We are therefore unfortunately unable to comply with this suggestion. 

I noticed the absence of a general conclusion for the study. Could you please provide a brief conclusion that encapsulates key findings and highlights elements pertinent for other researchers and field workers in your area, as well as in other regions globally?

We apologize for the oversight, Thank you for this important suggestion. A conclusion has been added. 

Figure 4.1. and 4.2.: what do the asterisks mean? Explain in the caption please.

“Significance codes: ‘***’ < 0.001, ‘**’ < 0.01, ‘*’ < 0.05, ‘.’ < 0.1‘” have been added to the captions of both figures.

Figure 4.2.: what is Vint and Vimp? Please, explain this in the caption.

These abbreviations are now being explained in the caption. Thanks for highlighting this. “Vint” stands for the importance of interactions and “Vimp” stands for the importance of variables (predictors). We added this information to the legend of Figure 4.2.

Reviewer #3: 

General comment:

The authors address an important topic and use adequate vocabulary to report their study. Perinatal mortality is a welfare issue and currently below target. I cannot assess all information due to very low resolution in the images (at least in the version I obtained). 

Thank you very much for your time and effort and the valuable comments, which helped us improve the manuscript. 

We provided high DPI images to the journal, so we assume the low resolution in the version you obtained must be an issue with data transfer following submission. We include the images at the very end of this response for your convenience.

Please note: When we mention line numbers this always refers to the initial submission, as line numbers have changed in the revised version due to numerous changes during the revision process.

Also I have a hard time understanding what happened during data collection and model building. I cannot assess the raw data quality. Also, I find it difficult to navigate through potential contradictions (for instance when it comes to the level of the analysis). There is little information on model architecture, at least the statistical equations should be printed in a paper whose aims include comparison of methods. It is unfortunate, that much space in the manuscript is used for repetition (see comments to discussion sections), whereas information for someone who reads your paper for the first time lacks. Please revise. 

Information on the raw data quality has been added – all raw data was recorded by the farmers. Discussion of this matter has now also been added. 

We have taken your comments into account and have added the requested information, and have revised the discussion accordingly. 

We added several clarifications for methods and data quality to the M&M section and answered all your questions to the best of our ability. 

We also added “cow-level” to the title, abstract and introduction to clarify the level of the analyses.

We agree that providing equations is important. We are now providing these in a supplementary file (supplementary material) to be published alongside the main paper, with the details of the equations for the multivariable model without interactions and for the final model with 4 interactions. Incorporating these equations into the main text, particularly the latter model with interactions, would greatly hinder readability because it is very big (>30 coefficients).

The latter model is the result of the brute-force variable (or model) selection, so there is no extra formula from the brute-force approach. We could not extract any equation from the Random Forest model which determined variable importance. We have however added detailed information on the modelling approaches (R code) to the supplemental file. 

L 18, 44: There are no necessary calf losses, please delete ‘unnecessary’.

We agree – the term has been removed.

L 20: Can knowledge be comparable? Please revise style.

This sentence has been re-phrased.

L 33: I reckon you investigated associations, not causalities. Please consider revising “influential factors”.

Thank you for highlighting this. Yes, we did study associations. In addition, we also studied the variable importance by several methods. We therefore re-phrased this as “most important factors”.

L38: Your abstract lacks two or three discussion-like sentences concerning a) veterinary content and b) statistical methodological findings. This is also a problem in the discussion section itself, where interpretation is lacking. 

We have added discussion on both topics to the abstract. More interpretation has also been added to the discussion section.

L92 Please add that Saxony and Thuringia are political units/regions/provinces if this is the case and relevant. 

We added “federal states of Saxony and Thuringia”.

L 115: Classic log reg may be misleading in a multi-farm setting, not mixed log reg. Please carefully distinguish the model names as here it is unclear which model use mean (classic vs. log reg).

We have revised this sentence to clarify.

Additionally, as recommended by another reviewer, we moved this part to M&M. 

L 131: Please provide a detailed sample size calculation here, including software used and name of test.

Details of the calculation of the sample size have been added to M&M.

L136: We haven’t learnt about regional division of the study team yet. Why didn’t you use the national farm registration database in Bavaria?

It was attempted to use the national farm registration database in all regions. However, due to data protection legislation, the researchers did not have direct access to the contact details of the farmers via the data base and instead had to rely on the co-operation of the organizations which held these access rights to personal details, such as regional veterinary authorities. Due to the federal structure of the country, each federal state has their own veterinary authority. While these authorities agreed to act as a point of contact to the farmers in regions North and East, they refused in Bavaria. The researchers therefore had to resort to the milk recording database (with covers approximately 90% of all dairy cattle in that state) and the milk recording association (Milchprüfring Bayern e.V.) for farm selection and as a point of contact with the farmers. Details of this recruitment process are given in the comprehensive final report of the underlying cross-sectional study, which is cited in the text so the interested reader can refer to it for further detail. A brief explanation of this issue has also been added to the text.

L138: How was this response rate distributed across regions? It is not convenient to go to other articles to understand crucial information of the present one. 

This section has been re-phrased to add the required information. We have also identified and corrected an error in the initial text, participation ranged between 5.9% and 14.5% of invited farms in the different regions.

L143: How many researchers were there in total and per region? I acknowledge that you need various staff members to cover the farms across regions. A more detailed insight into potential observation biases will be helpful.

The number of researchers per region has been added – this was a very large project that covered a large variety of husbandry and health indicators, hence the large number of researchers involved with the various aspects of the study – the data presented here form just a very small part of the underlying cross-sectional study. As described in the M&M section, the calving records are farmer-recorded, retrospective data that were made available to the project by the farmers (to cover the 12 months prior to the farm visit). Observation bias for this data therefore lies with the farmers rather than the researchers, who were merely granted access to these records from the compulsory farm recording system by the participating farmers. We hope the changes we made to the text clarify this issue. We have also added discussion of the topic of farmer-recorded data to the discussion. 

L153: Please briefly describe this procedure here. 

We added a brief description of the procedure. 

L155: This is a result and needs to go to the results sections. How many farmer responded and how many were excluded?

L155ff: dito

This information had already been stated in the initial submission, albeit a bit scattered across the M&M section: Of the 765 participating farms in the underlying study, 44 could not provide some of the data required for this analysis and were thus excluded, leaving 721 farms to contribute to the present study. We have now brought this information together in the same section of the text for clarity. We however think this information is still part of data acquisition and data handling, leading to the creation of the final dataset for this study, and have therefore opted to leave it in the material and methods section rather than moving it to results. 

L 174: Provide information on gestation length data quality in order to be able to follow your approach.

We do not have information on the insemination dates corresponding to the studied parturitions for a large proportion of the studied cases. The statement “insemination dates were poorly recorded” was therefore incorrect and misleading, we apologize for this mistake and have since re-phrased to clarify. They were simply not available for many animals from the given dataset. That is because the available data set only covered a 12-month period for each farm. Gestation length could have only been calculated for those cows for which both the insemination and corresponding calving dates fell within the 12 month period. This is naturally a very small proportion of the studied animals. 

We have now added a statement to clarify that this study made retrospective use of an existing data set that was initially collected with other scientific topics in mind. The current authors would definitely have planned a different approach to data collection and to the information to be collected, had they been able to prospectively plan this study.

L197: Please add seasons’ cutoffs in months or weeks. 

This information has been added. 

L207: What did qualify a variable to be offered to the model? Did you perform univariate analysis first or did you add all variables per se?

Yes, we did perform univariable analysis first. (Fig 4.1a “Univariable models”). Only season was not significant univariably (p = 0.27). But there are 2 reasons we kept all 5 predictors including “season” in the multivariable model.

 The model without season did not significantly improve the quality of the model:

 since we only had 5 predictors and since season became part of 2 out of 4 important interactions, season remained part of the multivariable model.

L207: Did your models fail to converge when not using BOBYQA optimizer?

Yes. Here is the warning we get when we use the default ”Nelder_Mead” optimizer: 

L207: Did you check model assumptions for logistic models besides multicollinearity? How did you handle outliers? Did you check linearity of the logit for those variables that were continuous?

Yes, we checked for influential observations via residuals. And while there are some “red” points on the plot below, the points stay inside the contour lines. Moreover, since all 5 of our predictors are categorical, we are not sure outliers in the data (instead of residuals) can be found.

Similarly, since all 5 of our predictors are categorical, and none are continuous, we think the linearity of the logit assumption cannot be checked. You also explicitly mentioned “for those variables that were continuous”. There were no continuous variables. 

L209: Which factors did you consider to be possible confounding factors? Did you offer all interaction combinations as default? It is not clearly state in my opinion and thus hard to follow.

We are sorry, we meant “risk factors”, not “confounding factors”. This has been changed accordingly.

Following your comment, we added clarification on the brute-force approach: Specifically, we evaluated 32 models resulting from various combinations of five predictors and an additional 1450 models generated from all possible pairwise interactions between these predictors. The “brute-force” approach ranked these models based on their Akaike Information Criterion (AIC) values, identifying the optimal model or a set of qualitatively similar models within a 2-unit range of AIC. Notably, this approach allowed us to identify the best-performing model, characterized by four interactions (as depicted in Figure 4.3), with the corresponding model equation provided in the supplementary materials. Subsequently, we compared this model’s AIC with that of the “final” model obtained through a backwards-selection approach, aiming to contrast the two variable selection methodologies.

L225: Did you mention that you conducted your analysis at the cow-level any earlier than that? I believe you did not. Please specify this in introduction and abstract. I also suggest to specify this in the title. 

You are right, we apologize for that. We corrected this oversight and have now added this information to the title, abstract, introduction and other parts throughout the manuscript. 

You use also non-cow-level such as season. This leaves me somewhat puzzled as to which is the level of the model. I understand that you account for the hierarchical structure of the data at the farm level. Please clarify.

We have now clarified that the study was undertaken at the cow level. “Season” will also affect each individual cow (and her newborn calf) at the time she gives birth. Previous studies have also used “season” when analyzing cow-level data. 

 L256: …with different methods than what?

We clarified in the text: “with four different methods in this study”.

L259: Earlier you state that models without interaction terms are more prone to confounding. 

We are sorry for the confusion. This sentence was deleted, because we actually applied 8 different modelling-settings, and not just 8 models. For example, we now added more information to the M&M, e.g.: (1) five univariable mixed-effects models, (2) multivariable model without interactions, (3) ten separate bivariable pairwise interactions models and (3) final model with multiple only relevant interactions. The section describing brute-force has also been adjusted and now describes how many models were created. The M&M section has also been slightly re-arranged due to the request by another reviewer, we hope it is now easier to follow.

Why did you now do model without interaction terms? This is not clear to me. Compare with L 3034.

We had initially stated that one of the purposes of the study was the comparison of different statistical approaches (e.g. univariable, multivariable without and multivariable with interactions, RF and brute-force) in order to compare the results among these approaches, and most importantly - to allow for a better comparison of our results to past and future studies. We truly believe that there is not a single best or correct model or approach, but that a multifaceted statistical analysis is less subjective and less biased, while at the same time more comparable and produces more inference.

Thus, we present the results of both univariable and multivariable models (without interactions first) but interpret only multivariable results. We present the model without interactions, because most of our colleagues do not use interactions, so they won’t be able to compare the results of their studies with the same predictor to ours. 

The manuscript has been revised to provide a clearer line of thought, and we hope this clarifies this issue.

L262: The variable breed is strongly unequally distributed among categories. Please discuss to what extent this may have biased your results and how you accounted for it. Does this effect comparability within your study, i.e., do you think it is appropriate to compare data from ~115,000 GH to ~1,300 BS?

Thank you for highlighting this issue – we had not mentioned that all model predictions (of any of our models) and contrasts were calculated with the proportional weights, where the model accounts for the proportion of the frequencies of the categories (in the original data) that are averaged over. This information has now been added to the M&M. Since we accounted for it, there is no such bias in our results. 

L 263: Please provide mean and sd for PM across farms. 

This information has been added.

L 268: dito concerning distribution of the data among categories. How did you handle (relative) low numbers in calving ease? How did this influence model building?

Proportional weight as mentioned above.

L 285-288: Please adhere to scientific English rather than colloquial English through use of parallel structure, i.e., name variables which are significantly associated first for better readability. 

This has been corrected.

Fig 4.1.: The low resolution of the figures is inconvenient, especially in Fig 4.2. I do not know if this happened automatically/unintentionally when submitting or is a bug from the submission tool. I cannot read it and therefore cannot assess it. 

We are sorry you did not receive the images in good enough quality. We submitted high resolution images to the Journal, so we assume this might indeed be a bug from the submission tool. As mentioned above, we added the images at the end of this file for your convenience and we hope you receive them now in better quality. 

L303-307: This is unclear to me. Please explain. Also, do you mean ‘predictors’ or ‘potential predictors’?

We mean predictors, not potential predictors. Multivariable models are more realistic than univariable analyses. But univariable models (or tests) are still often used and will be used in the future. So, we present the results of both uni- and multivariable models. The univariable results are presented for comparison. In a multivariable model the predictors “coexist” in their influence on the outcome. For the assessment of the influence of one predictor on PM, the other predictors need to be held constant. Constant at average for the other numeric predictors (which we don’t have) and often at the first category for categorical predictors. So, having 5 categorical predictors, when the model keeps the 4 of them constant at the very first category, then we will get a very different result, like primiparous for parity, small farms for farm-size etc. because they are the first categories in categorical predictors. Therefore, in our case the “adjustment” we meant and wanted that predictions for let’s say “breed” will be averaged over primi-, secondi- and multiparous for the case of parity (instead of keeping the parity constant at only primiparous) or averaged over all sizes of farms (instead of keeping farm-size constant at the first – namely – small farms category). We hope we could answer your question.

L323-324: How did you calculate this? Please provide guidance, a code snippet or explanations on this. 

Below (and in the supplementary materials), using the “at” argument in the “emmeans” function we can ask for any category from any predictor. And since we already knew that primiparous cows have higher probability of PM, that large farms have higher probability etc., we could ask our model to assess the probabilities for the combination of predictors which would result in the “worst case scenario”. The “best case scenario” is calculated in the same manner. 

Table 4.2.: What is the advantage of combining predictors (at the cow-level)? It reads as if a cow you be attributed to two breeds or as if you grouped calvings of an individual pluriparous cow. Would you please elaborate on this? Are the ‘/’ indicating reference levels? There is information missing to understand your results when going through the article with a perspective from a first reader. 

The table legend and the table itself have been modified to clarify that this table reports pairwise contrasts, not combinations of predictors.

First, we added the “on cow-level” to the table legend to clarify. Secondly, “/” indicates ratio, for Odds Ratios. And since odds-ratios are usually a reference level divided by one of the other levels, it is written in this way “reference/non_reference. Detailed explanation has been added to the legend of tables 4.2., 4.3. and 4.4.

L 453f: Are these absolute or relative percentage values? Where did you report this data in the manuscript?

These are absolute predicted probabilities from the total sample size (reported at Line 160 of the initial submission) which was calculated from the univariable model only:

L527: I understand your argumentation, however, have a hard time believing that any risk factor applies to all individuals of all populations. To me, your sentence reads like this. I would appreciate you weakened that statement to something alike “not an important/predominant predictor”. 

We have changed this statement accordingly.

General remark to the discussion: I miss a discussion on data quality of the raw data. Do you think there is little entry errors when it comes to parity or calving ease? What about inter- and intraobserver reliability when there is hundreds of farmers involved? I appreciate your huge dataset but would be more cautious when interpreting this data. How is data entered into the system? What is the maximal delays for data entry?

We apologize for this oversight. This is indeed an important topic. Discussion on data quality of the raw data has been added. More information has also been added on data recording and timescale (maximal delays) by the farmers. 

L581f: This should be based on suitable literature as it reads somewhat as a speculation.

We have added references here. 

L591: You shouldn’t be repeating your results and methods during discussion. This is actually making your paper lengthy and blurs new information among already given information. This is a general style remark also applying to earlier parts of the discussion.

Thank you for this helpful comment. Extensive reference to results or previous publications has now been removed from the discussion. The discussion has also been partly re-arranged, re-phrased and re-focused to include clinical relevance of the results (see below) 

L608: dito

See above.

L643: dito. Please revise the discussion section and put emphasis on what your results mean to the reader, to the cows, to the future (of PM) etc and refrain from giving the same information from M&M or results in other words. Should we prioritize SIM breeds over GH in those German regions now, due to increased PM? Is the trade-off between a likely higher average milk yield and more dead calves acceptable? If yes, why? Should we transition to seasonal calvings to summer? If not, why? What type of programs could be deployed to minimize PM? As experts in the field, do you think that there is practical approaches beyond theory that may work? 

Thank you for this important comment – Discussion on these clinical topics and practical implications of the results has been added.

L 642/L205/255ff: The level of analysis is farm and not animal or both? Why didn’t you include a random effect for animal? Didn’t your dataset offer several calvings from the same individual?

This is a fair concern. However, since every farm only provided a 12-month dataset, multiple calvings from the same individual within these 12 months were extremely rare. We have added the number of animals contributing to the calving records to the beginning of the results section to clarify. In fact, 98.4% of the cows did have only one calving record, so there were not enough repeated measures for “individual cow” for us to include this as a random effect. But, out of curiosity, we actually created a model with nested random effect (1|farm_id/animal_id) and this model did not significantly improve the quality of the model (which we also expected, since 98.4% of animal_IDs are not repeated): (a statement has been added to the M&M section)

L681: For me as an intermediate in veterinary statistics, it is not comprehensible from your explanations how you transitioned from unbalanced to the balances RF model. I think the great majority of your readers would appreciate you elaborated on this in more detail.

Based on your questions so far, your stats knowledge is way above intermediate vet scientist. Which we greatly appreciate! 

There is no transition. We added a more detailed explanation about what “balanced” and “not balanced” RF mean, including examples. For instance, not balanced RF would classify 94% survived vs. 6% not-survived calves, while balanced RF would classify 50% survived vs. 50% not-survived, where we resampled 1000 PM and 1000 not-PM datapoints for every of 10000 trees. We also added that such balance is achieved via down-sampling the majority class (survived), while over-sampling the minority class (dead). The not-balanced RF would simply predict that 100% of calves survive, and it would be 94% correct. We also added a reference which originally described this problem. 

In our experience, not balanced RF always produces the results different from GLMMs, and we think it’s because it maximizes the prediction accuracy. So, predicting “survival” (no PM) is a “save bet” for RF because most of the calves survived. And we were actually pleasantly surprised that balanced RF (even without any random effect), produced similar results to GLMMs.

L685: Backwards selection does not work through adding of variables. Please proof me wrong or delete.

Sorry, we meant both backwards and forwards, this has been re-phrased.

---

## [Decision Letter · Decision Letter 1]

27 Mar 2024

Perinatal mortality in German dairy cattle: unveiling the importance of cow-level risk factors and their interactions using a multifaceted modelling approach.

PONE-D-23-30463R1

Dear Dr. Zablotski,

We’re pleased to inform you that your manuscript has been judged scientifically suitable for publication and will be formally accepted for publication once it meets all outstanding technical requirements.

Kind regards,

Angel Abuelo, DVM, MRes, MSc, PhD, DABVP (Dairy), DECBHM

Academic Editor

PLOS ONE

Additional Editor Comments (optional):

Reviewers' comments:

Reviewer's Responses to Questions

**Comments to the Author**

1. If the authors have adequately addressed your comments raised in a previous round of review and you feel that this manuscript is now acceptable for publication, you may indicate that here to bypass the “Comments to the Author” section, enter your conflict of interest statement in the “Confidential to Editor” section, and submit your "Accept" recommendation.

Reviewer #1: All comments have been addressed

Reviewer #2: All comments have been addressed

Reviewer #3: All comments have been addressed

2. Is the manuscript technically sound, and do the data support the conclusions?

Reviewer #1: (No Response)

Reviewer #2: Yes

Reviewer #3: Yes

3. Has the statistical analysis been performed appropriately and rigorously? 

Reviewer #1: Yes

Reviewer #2: Yes

Reviewer #3: Yes

4. Have the authors made all data underlying the findings in their manuscript fully available?

Reviewer #1: Yes

Reviewer #2: Yes

Reviewer #3: Yes

5. Is the manuscript presented in an intelligible fashion and written in standard English?

Reviewer #1: Yes

Reviewer #2: Yes

Reviewer #3: Yes

6. Review Comments to the Author

Reviewer #1: The reviewer thanks the author for answering the suggestions/comments properly. The manuscript is now more concise.

Figure 4.1. If the units are log-odds describe that in the caption of the figure.

Reviewer #2: Many thanks for the thorough revision of the manuscript and for your contribution to the understanding of PM.

Reviewer #3: (No Response)

7. PLOS authors have the option to publish the peer review history of their article (what does this mean?). If published, this will include your full peer review and any attached files.

Reviewer #1: No

Reviewer #2: No

Reviewer #3: No

---

## [Editor Report · Acceptance letter]

4 Apr 2024

PONE-D-23-30463R1 

PLOS ONE

Dear Dr. Zablotski, 

I'm pleased to inform you that your manuscript has been deemed suitable for publication in PLOS ONE. Congratulations! Your manuscript is now being handed over to our production team.

Kind regards, 

on behalf of

Dr. Angel Abuelo 

Academic Editor

PLOS ONE